# In-Context Denoising with One-Layer Transformers: Connections between Attention and Associative Memory Retrieval

**Matthew Smart** [1]  **Alberto Bietti** [2]  **Anirvan M. Sengupta** [2 3 4]

## Abstract

We introduce in-context denoising, a task that refines the connection between attention-based architectures and dense associative memory (DAM) networks, also known as modern Hopfield networks. Using a Bayesian framework, we show theoretically and empirically that certain restricted denoising problems can be solved optimally even by a single-layer transformer. We demonstrate that a trained attention layer processes each denoising prompt by performing a single gradient descent update on a context-aware DAM energy landscape, where context tokens serve as associative memories and the query token acts as an initial state. This one-step update yields better solutions than exact retrieval of either a context token or a spurious local minimum, providing a concrete example of DAM networks extending beyond the standard retrieval paradigm. Overall, this work solidifies the link between associative memory and attention mechanisms first identified by Ramsauer et al., and demonstrates the relevance of associative memory models in the study of in-context learning.

## 1. Introduction

The transformer architecture (Vaswani et al., 2017) has achieved remarkable success across diverse domains, from natural language processing (Devlin et al., 2019; Brown et al., 2020; Touvron et al., 2023) to computer vision (Dosovitskiy et al., 2021). Despite their practical success, understanding the mechanisms behind transformer-based networks remains an open challenge. This challenge is exacerbated by the growing scale and complexity of modern large networks. Toward addressing this, researchers studying simplified architectures have identified connections between the attention operation that is central to transformers and associative memory models (Ramsauer et al., 2021), providing not only an avenue for understanding how such architectures encode and retrieve information but also potentially ways to improve them further.

The most celebrated model for associative memories in systems neuroscience is the so-called Hopfield model (Amari, 1972; Nakano, 1972; Little, 1974; Hopfield, 1982). This model has a capacity to store "memories" (stable fixed points of a recurrent update rule) proportional to the number of nodes (Hopfield, 1982; Amit et al., 1985). In the last decade, new energy functions (Krotov & Hopfield, 2016; Demircigil et al., 2017) were proposed for dense associative memories with much higher capacities. These energy functions are often referred to as modern Hopfield models. Ramsauer et al. (2021) pointed out the similarity between the one-step update rule of a certain modern Hopfield network (Demircigil et al., 2017) and the softmax attention layer of transformers, generating interest in the statistical physics and systems neuroscience communities (Krotov & Hopfield, 2021; Krotov, 2023; Lucibello & Mézard, 2024; Millidge et al., 2022). Recent work has extended this concept to improve retrieval by incorporating sparsity (Hu et al., 2023; Wu et al., 2024b; Santos et al., 2024; Wu et al., 2024a), while others have leveraged associative memory principles to design new energy-based transformer architectures (Hoover et al., 2023). However, these extensions and the foundational construction in Ramsauer et al. (2021) primarily focus on the specific task of exact retrieval (converging to a fixed point), while in practice transformers may tackle many other tasks.

To explore this connection beyond retrieval, we introduce *in-context denoising*, a task that bridges the behavior of trained transformers and associative memory networks through the lens of in-context learning (ICL). In standard ICL, a sequence model is trained to infer an unknown function $g$ from contextual examples, predicting

---

[1] Center for Computational Biology, Flatiron Institute, New York, NY, USA [2] Center for Computational Mathematics, Flatiron Institute, New York, NY, USA [3] Center for Computational Quantum Physics, Flatiron Institute, New York, NY, USA [4] Department of Physics and Astronomy, Rutgers University, Piscataway, NJ, USA. Correspondence to: Matthew Smart <msmart@flatironinstitute.org>, Anirvan M. Sengupta <anirvans.physics@gmail.com>.

*Proceedings of the 42^{nd} International Conference on Machine Learning*, Vancouver, Canada. PMLR 267, 2025. Copyright 2025 by the author(s).

$g(X_{L+1})$ given a sequence of input-output pairs $E = ((X_1, g(X_1)), ..., (X_L, g(X_L)), (X_{L+1}, -))$. Crucially, $g$ is implied solely through the context and differs across prompts – performant models are therefore said to "learn $g(x)$ in context". While ICL has been extensively studied in supervised settings (Garg et al., 2022; Zhang et al., 2024; Akyürek et al., 2023; Reddy, 2024), recent work suggests that transformers may internally emulate gradient descent over a context-specific loss function during inference (Von Oswald et al., 2023; Dai et al., 2023; Ahn et al., 2023). This general perspective aligns with our findings.

In this work, we generalize ICL to an unsupervised setting where the prompt consists of $L$ samples from a random distribution and the query is a noise-corrupted sample from the same distribution. This shift allows us to probe how trained transformers internally approximate Bayes optimal inference, while deepening the connection to associative memory models which are prototypical denoisers. By setting up this problem in this way, we also attempt to answer a few questions. One concerns the memorization-generalization dilemma in denoising: a Hopfield model's success is usually measured by successful memory recovery, while in-context learning may have to solve a completely new problem. Another question has to do with the number of iterations of the corresponding Hopfield model: why does the Ramsauer et al. (2021) correspondence involve only one iteration of Hopfield energy minimization and not many?

**In summary, our contributions are as follows:** In Section 2, we introduce in-context denoising as a framework for understanding how transformers perform implicit inference beyond memory retrieval. In Section 3, we establish that single-layer transformers with one attention head are expressive enough to optimally solve certain denoising problems. We then empirically demonstrate that standard training from random weights can recover the Bayes optimal predictors. The trained attention layers are mapped back to dense associative memory networks in Section 4. Our results refine the general connection pointed out in previous work, offer new mechanistic insights into attention, and provide a concrete example of dense associative memory networks extending beyond the standard memory retrieval paradigm to solve a novel in-context learning task.

## 2. Problem formulation: In-context denoising

In this section, we describe our general setup. Recurring common notation is described in Appendix A.1.

### 2.1. Setup

Each task corresponds to a distribution $D$ over the probability distribution of data: $p_X \sim D$. Let $X_1, \cdots, X_{L+1} \overset{\text{iid}}{\sim} p_X$, define the sampling of the tokens. Let the noise cor-

ruption be defined by $\tilde{X} \sim p_{\text{noise}}(\cdot | X_{L+1})$. The random sequence $E = (X_1, X_2, ..., X_L, \tilde{X})$ are given as "context" (input) to a sequence model $F(\cdot; \theta)$ which outputs an estimate $\hat{X}_{L+1}$ of the original $(L+1)$-th token . The task is to minimize the expected loss $\mathbb{E}[l(\hat{X}_{L+1}, X_{L+1})]$ for some loss function $l(\cdot, \cdot)$. Namely, our problem is to find

$$\min_{\theta} \mathbb{E}_{p_X \sim D, X_{1:L+1} \sim p_X^{L+1}, \tilde{X} \sim p_{\text{noise}}(\cdot | X_{L+1})}[l(F(E, \theta), X_{L+1})]. \tag{1}$$

In practice, we choose $\tilde{X} = X_{L+1} + Z$, a pure token corrupted by the addition of isotropic Gaussian noise $Z \sim \mathcal{N}(0, \sigma_Z^2 I_n)$, and our objective function to minimize is the mean squared error (MSE) $\mathbb{E}[||\hat{X}_{L+1} - X_{L+1}||^2]$.

In the following subsection, we explain the pure token distributions for three specific tasks. These tasks are of course structured so that a one-layer transformer has the expressivity to capture a solution, which, as $L \to \infty$, provides an optimal solution, in some sense. To that end, we derive Bayes optimal estimators for each of the three tasks, under the assumption that we know the original distribution $p_X$ of pure tokens. In Section 3, we use these estimators as baselines to evaluate the performance of the denoiser $f(E, \theta)$ based on a one-layer transformer trained on finite datasets.

### 2.2. Task-specific token distributions

We consider three elementary in-context denoising tasks, where the data (vectors in $\mathbb{R}^n$) comes from:

1. Linear manifolds ($d$-dimensional subspaces)

2. Nonlinear manifolds ($d$-spheres)

3. Small noise Gaussian mixtures (clusters) where the component means have fixed norm

Below we describe the task-specific distributions $p_X$ and the process for sampling tokens $\{x_t\}$. The same corruption process applies to all cases: $\tilde{X} = X_{L+1} + Z, Z \sim \mathcal{N}(0, \sigma_Z^2 I_n)$.

#### 2.2.1. CASE 1 - LINEAR MANIFOLDS

A given training prompt consists of pure tokens sampled from a random $d$-dimensional subspace $S$ of $\mathbb{R}^n$.

- Let $P$ be the orthogonal projection operator to a random $d$-dim subspace $S$ of $\mathbb{R}^n$, sampled according to the uniform measure, induced by the Haar measure on the coset space $O(n)/O(n-d) \times O(d)$, on the Grassmanian $G(d, n)$, the manifold of all $d$-dimensional subspaces of $\mathbb{R}^n$.

- Let $Y \sim \mathcal{N}(0, \sigma_0^2 I_n)$ and define $X = PY$; we use this procedure to construct the starting sequences $(X_1, ..., X_{L+1})$ of $L+1$ independent tokens.

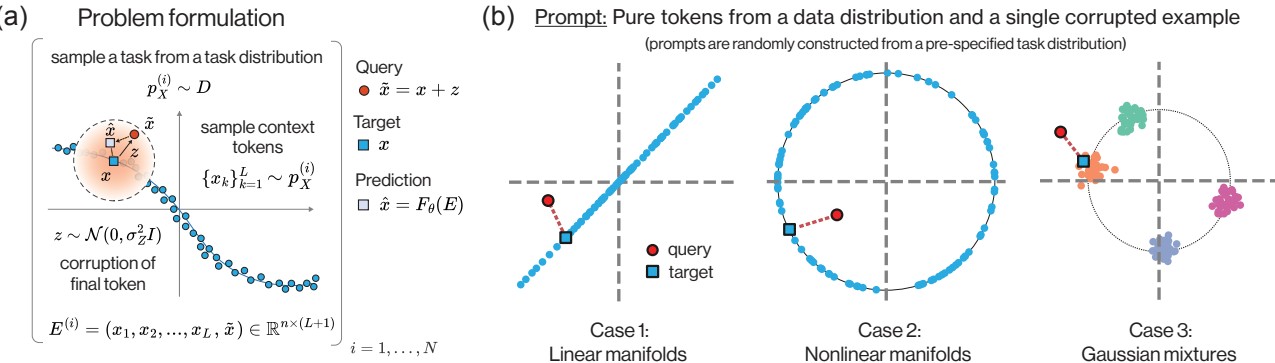

*Figure 1.* (a) Problem formulation for a general in-context denoising task. (b) The three denoising tasks considered here include instances of linear and non-linear manifolds as well as Gaussian mixtures. In each case, the task embedding $E^{(i)}$ consists of a sequence of pure tokens from the data distribution $p_X^{(i)} \sim D$ where $D$ denotes the task distribution, along with a single query token that has been corrupted by Gaussian noise. The objective is to predict the target (i.e. *denoise* the query) given information contained only in the prompt.

We thus have $p_X = \mathcal{N}(0, \sigma_0^2 P)$, with the Haar distribution of $P$ characterizing the task ensemble associated with $D$.

### 2.2.2. CASE 2 - NONLINEAR MANIFOLDS

We focus on the case of $d$-dimensional spheres of fixed radius $R$ centered at the origin in $\mathbb{R}^n$.

- Choose a random $d+1$-dimensional subspace $V$ of $\mathbb{R}^n$, sampled according to the uniform measure, as before, on the Grassmanian $G(d+1, n)$. The choice of this random subspace generates the distribution of tasks $D$.

- Inside $V$, sample uniformly from the radius $R$ sphere (once more, a Haar induced measure on a coset space $O(d+1)/O(d)$). We use this procedure to construct input sequences $X_{1:L+1} = (x_1, ..., x_{L+1})$ of $L+1$ independent tokens.

In practice, we uniformly sample points with fixed norm in $\mathbb{R}^d$ and embed them in $\mathbb{R}^n$ by concatenating zeros. We then rotate the points by selecting a random orthogonal matrix $Q \in \mathbb{R}^{n \times n}$.

### 2.2.3. CASE 3 - GAUSSIAN MIXTURES (CLUSTERING)

Pure tokens are sampled from a weighted mixture of isotropic Gaussians in $n$-dimensions, $\{w_a, (\mu_a, \sigma_a^2)\}_{a=1}^K$. The density is

$$p_X(x) = \sum_{a=1}^K w_a C_a e^{-\|x - \mu_a\|^2/2\sigma_a^2},$$

where $C_a = (2\pi\sigma_a^2)^{-n/2}$ are normalizing constants. The $\mu_a$ are independently chosen from a uniform distribution on the radius $R$ sphere of dimension $n-1$, centered around

zero. The distribution of tasks $D$, is decided by the choice of $\{\mu_a\}_{a=1}^K$.

For our ideal case, we will consider the limit that the variances go to zero. In that case, the density is simply

$$p_{X_0}(x) = \sum_{a=1}^K w_a \delta(x - \mu_a).$$

### 2.3. Bayes optimal denoising baselines for each case

The first $L$ tokens in $E$ are "pure samples" from $p$ that should provide information about the distribution for our denoising task. Our performance is expected to be no better than that of the best method, in the case that the token distribution and also the corrupting process are exactly known. This is where the Bayesian optimal baseline comes in. As is well-known, the Bayes optimal predictor of a quantity is given by the posterior mean. We use that fact to compute the Bayes optimal loss.

In particular, we seek a function $f : \mathbb{R}^n \to \mathbb{R}^n$ such that $\mathbb{E}_{X,\tilde{X}} \left[ \|X - f(\tilde{X})\|^2 \right]$ is minimized. Since the perturbation $Z$ is Gaussian, the posterior distribution of $X$, given $\tilde{X}$ is

$$p_{X|\tilde{X}}(x \mid \tilde{x}) = C(\tilde{x}) p_X(x) e^{-\|x - \tilde{x}\|^2/2\sigma_Z^2},$$

where $C(\tilde{x})$ is a normalizing factor (see Appendix A.2 for more explanation). The following proposition sets up a baseline to which we expect to compare our results as $L \to \infty$. The proof is in Appendix B.1.

**Proposition 1.** *For each task, specified by the input distribution $p_X$, and the noise model $p_{\tilde{X}|X}$,*

$$\mathbb{E}_{X,\tilde{X}} \left[ \|X - f(\tilde{X})\|^2 \right] \geq \mathbb{E}_{\tilde{X}} \left[ \mathrm{Tr}\, \mathrm{Cov}(X \mid \tilde{X}) \right]. \quad (2)$$

*This lower bound is met when $f(\tilde{X}) = \mathbb{E}[X \mid \tilde{X}]$.*

Thus, the Bayes optimal denoiser is the posterior expectation for $X$ given $\tilde{X}$. The expected loss is found by computing the posterior sum of variances.

These optimal denoisers can be computed analytically for both the linear and nonlinear manifold cases (given the variances and dimensionalities). In the Gaussian mixture (clustering) case, it depends on the choice of the centroids which then needs to be averaged over.

**Linear case.** For the linear denoising task, pure samples $X$ are drawn from an isotropic Gaussian in a restricted subspace. The following result provides the Bayes optimal predictor in this case, the proof of which is in Appendix C.1.

**Proposition 2.** *For $p_X$ corresponding to Subsection 2.2.1, the Bayes optimal answer is*

$$f_{opt}(\tilde{X}) = \mathbb{E}[X|\tilde{X}] = \frac{\sigma_0^2}{\sigma_0^2 + \sigma_Z^2} P\tilde{X}, \qquad (3)$$

*and the expected loss is*

$$\mathbb{E}\left[\|P\tilde{X} - X_{L+1}\|^2\right] = d\sigma_0^2 \sigma_Z^2 / (\sigma_0^2 + \sigma_Z^2). \qquad (4)$$

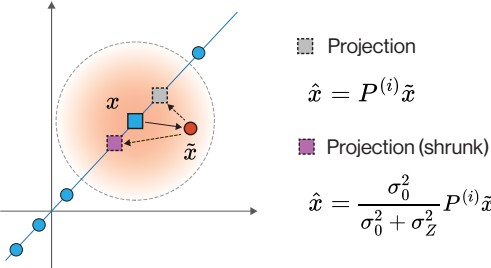

*Figure 2.* Baseline estimators for the case of random linear manifolds with projection operator $P^{(i)}$.

**Manifold case.** In the nonlinear manifold denoising problem, we focus on the case of lower dimensional spheres $S$ (e.g. the circle $S^1 \subset \mathbb{R}^2$). For such manifolds, the Bayes optimal answer is given by the following proposition.

**Proposition 3.** *For $p_X$ defined as in Subsection 2.2.2, with $P$ being the orthogonal projection operator to $V$, the $d + 1$ dimensional linear subspace, with $R$ being the radius of sphere $S$, the Bayes optimal answer is*

$$f_{opt}(\tilde{X}) = \mathbb{E}[X \mid \tilde{X}]$$

$$= \frac{\int e^{\langle x, \tilde{X}_\parallel \rangle / \sigma_Z^2} \, x \, dS_x}{\int e^{\langle x, \tilde{X}_\parallel \rangle / \sigma_Z^2} \, dS_x} \qquad (5)$$

$$= \frac{I_{\frac{d+1}{2}}\left(R\frac{\|\tilde{X}_\parallel\|}{\sigma_Z^2}\right)}{I_{\frac{d-1}{2}}\left(R\frac{\|\tilde{X}_\parallel\|}{\sigma_Z^2}\right)} R\frac{\tilde{X}_\parallel}{\|\tilde{X}_\parallel\|}, \qquad (6)$$

*where $\tilde{X}_\parallel = P\tilde{X}$ and $I_\nu$ is the modified Bessel function of the first kind.*

**Clustering case.** For clustering with isotropic Gaussian mixtures $\{w_a, (\mu_a, \sigma_a^2)\}_{a=1}^p$, the Bayes optimal predictors for some important special cases are as follows. See Appendix C.3 for the general case.

**Proposition 4.** *For general isotropic Gaussian model with $\sigma_a = \sigma_0, \|\mu_a\| = R$ for all $a = 1, \ldots, K$.*

$$f_{opt}(\tilde{X}) = \mathbb{E}[X|\tilde{X}]$$

$$= \frac{\sigma_0^2}{\sigma_0^2 + \sigma_Z^2} \tilde{X} + \frac{\sigma_Z^2}{\sigma_0^2 + \sigma_Z^2} \frac{\sum_a w_a e^{\langle \mu_a, \tilde{X} \rangle / (\sigma_0^2 + \sigma_Z^2)} \, \mu_a}{\sum_a w_a e^{\langle \mu_a, \tilde{X} \rangle / (\sigma_0^2 + \sigma_Z^2)}}. \qquad (7)$$

*If $\sigma_0 \to 0$,*

$$f_{opt}(\tilde{X}) = \mathbb{E}[X \mid \tilde{X}] = \frac{\sum_a w_a e^{\langle \mu_a, \tilde{X} \rangle / \sigma_Z^2} \, \mu_a}{\sum_a w_a e^{\langle \mu_a, \tilde{X} \rangle / \sigma_Z^2}}. \qquad (8)$$

In all three cases, we notice similarities between the form of the Bayes optimal predictor, and attention operations in transformers, a connection which we explore below.

## 3. In-context denoising with one-layer transformers – Empirical results

In this section, we provide simple constructions of one-layer transformers that approximate (and under certain conditions, exactly match) the Bayes optimal predictors above.

**Input:** Let $p_X^{(1)}, \ldots, p_X^{(N)} \overset{\text{iid}}{\sim} D$, be distributions sampled for one of the tasks. For each distribution $p_X^{(i)}$, we sample $E^{(i)} := (X_1^{(i)}, \ldots, X_L^{(i)}, \tilde{X}^{(i)})$ taking value in $\mathbb{R}^{n \times (L+1)}$ be an input to a sequence model. We also retain the true $(L+1)$-th token $X_{L+1}^{(i)}$ for each $i$.

**Objective:** Given an input sequence $E^{(i)}$, return the uncorrupted final token $X_{L+1}^{(i)}$. We consider the mean-squared error loss over a collection of $N$ training pairs, $\{E^{(i)}, X_{L+1}^{(i)}\}_{i=1}^N$,

$$C(\theta) = \sum_{i=1}^N \|F(E^{(i)}, \theta) - x_{L+1}^{(i)}\|^2, \qquad (9)$$

where $F(E^{(i)}, \theta)$ denotes the parametrized function predicting the target final token based on input sequence $E^{(i)}$.

### 3.1. One-layer transformer and the attention between the query and pure tokens

To motivate our choice of architecture, let us start by discussing the linear case.

There we have $f_{\text{opt}}(\tilde{X}) = \frac{\sigma_0^2}{\sigma_0^2 + \sigma_Z^2} P\tilde{X}$. Note that, by the strong law of large numbers, $\hat{P} = \frac{1}{\sigma_0^2 L}\sum_{t=1}^L X_t X_t^T$ is a random matrix that almost surely converges component-by-component to the orthogonal projection $P$ as $L \to \infty$, since, for each $t$, $X_t X_t^T$ has the expectation $\sigma_0^2 P$ and that $X_t$ is a Gaussian random variable with zero mean and a finite covariance matrix. So we could propose

$$f(\tilde{X}) = \frac{\sigma_0^2}{\sigma_0^2 + \sigma_Z^2}\hat{P}\tilde{X} = \frac{1}{(\sigma_0^2 + \sigma_Z^2)L}\sum_{t=1}^L X_t\langle X_t, \tilde{X}\rangle. \tag{10}$$

We now consider a simplified one-layer linear transformer (see Appendices D.1 and D.2 for more detailed discussions) which still has sufficient expressive power to capture our finite sample approximation to the Bayes optimal answer. We define

$$\hat{X} = F_{\text{Lin}}(E, \theta) := \frac{1}{L}W_{PV}X_{1:L}X_{1:L}^T W_{KQ}\tilde{X} \tag{11}$$

taking values in $\mathbb{R}^n$, where $X_{1:L} := [X_1, \ldots, X_L]$ taking values in $\mathbb{R}^{n\times L}$, with learnable weights $W_{KQ}, W_{PV} \in \mathbb{R}^{n\times n}$ abbreviated by $\theta$. Note that, when $W_{PV} = \alpha I_n, W_{KQ} = \beta I_n$, and $\alpha\beta = \frac{1}{\sigma_0^2 + \sigma_Z^2}$, $F(E, \theta)$ should approximate the Bayes optimal answer $f_{\text{opt}}(\tilde{X})$ as $L \to \infty$. For a detailed discussion of the convergence rate, see Appendix E, in general, and Proposition 5, in particular.

Similarly, we could argue that the second two problems, the $d$-dimesional spheres and the $\sigma_0 \to 0$ zero limit of the Gaussian mixtures could be addressed by softmax attention

$$\hat{X} = F(E, \theta) := W_{PV}X_{1:L}\text{softmax}(X_{1:L}^T W_{KQ}\tilde{X}) \tag{12}$$

taking values in $\mathbb{R}^n$. The function $\text{softmax}(z) := \frac{1}{\sum_{i=1}^n e^{z_i}}(e^{z_1}, \ldots, e^{z_n})^T \in \mathbb{R}^n$ is applied column-wise.

For both problems, namely the spheres and the $\sigma_0 \to 0$ Gaussian mixtures, we could have $W_{PV} = \alpha I_n, W_{KQ} = \beta I_n$ with $\alpha = 1, \beta = 1/\sigma_Z^2$ providing Bayes optimal answers as $L \to \infty$.

In fact, we could make a more general statement about distributions $p_X$ where the norm of $X$ is fixed.

**Theorem 3.1.** *If we have a task distribution $D$ so that the support of each $p_X$ is the subset of some sphere, centered around the origin, with a $p_X$-dependent radius $R$, then the function*

$$F(( \{X_t\}_{t=1}^L, \tilde{x}), \theta^*) = \frac{\sum_{t=1}^L X_t e^{\langle X_t, \tilde{x}\rangle/\sigma_Z^2}}{\sum_{t=1}^L e^{\langle X_t, \tilde{x}\rangle/\sigma_Z^2}} \tag{13}$$

*converges almost surely to the Bayes optimal answer $f_{\text{opt}}(\tilde{x})$ for all $\tilde{x} \in \mathbb{R}^n$, as $L \to \infty$. The optimal parameter $\theta^*$ refers to $W_{PV} = I_n, W_{KQ} = \frac{1}{\sigma_Z^2}I_n$.*

The proof of the theorem is in Appendix D.3. See Appendix E, particularly Proposition 6, for consideration of convergence rates. Note that the condition of $p_X$ being supported on a sphere is not artificial as, in many practical transformers, pre-norm with RMSNorm gives you inputs on the sphere, up to learned diagonal multipliers.

Note that the natural form of attention that is suggested by our formulation of in-context denoising would involve Gaussian kernels:

$$\hat{X} = F_G(E, \theta) := \frac{\sum_t W_{PV}X_t e^{-\frac{1}{2}||W_K X_t - W_Q\tilde{X}||^2}}{\sum_t e^{-\frac{1}{2}||W_K X_t - W_Q\tilde{X}||^2}}. \tag{14}$$

The relation between softmax attention and the Gaussian kernel has been noted in (Choromanski et al., 2021; Ambrogioni, 2024) and a Gaussian kernel-based attention is implemented in (Chen et al., 2021). A related Hopfield energy, with $W_K$, $W_Q$, and $W_{PV}$ proportional to identity matrices, is proposed in (Hoover et al., 2024a).

For the linear case, we use linear attention, but that may not be essential. Informally speaking, the softmax attention model has the capacity to subsume the linear attention model.

**Proposition 3.2.** *As $\epsilon \to 0$,*

$$F\left(E, \left(\frac{1}{\epsilon}W_{PV}, \epsilon W_{KQ}\right)\right) = \frac{1}{\epsilon}W_{PV}\bar{X}$$
$$+ \frac{1}{L}W_{PV}\sum_{t=1}^L X_t(X_t - \bar{X})^T W_{KQ}\tilde{X} + O(\epsilon), \tag{15}$$

*where $\bar{X} = \frac{1}{L}\sum_{t=1}^L X_t$ is the empirical mean.*

See Appendix F for the details of small $W_{KQ}$ expansion and Appendix F.1 for the proof of Proposition 3.2.

For case 1, note that $\mathbb{E}[X_t] = 0$ and covariance of $X_t$ is finite, $E[\bar{X}] = 0$, and $E[||\bar{X}||^2] = O(\frac{1}{L})$, allowing us to drop $\bar{X}$ as $L \to \infty$. If, in addition, $\epsilon$ is small, only the second term survives. Thus, $F\left(E, \left(\frac{1}{\epsilon}W_{PV}, \epsilon W_{KQ}\right)\right)$ starts to approximate $F_{\text{Lin}}\left(E, (W_{PV}, W_{KQ})\right)$ when $L$ is large and $\epsilon$ is small, with $\epsilon\sqrt{L}$ large. We therefore could use the softmax model for all three cases.

### 3.2. Case 1 – Linear manifolds

The Bayes optimal predictor for the linear denoising task from Section 2.3 suggests that the linear attention weights should be scaled identity matrices with their product satisfying $\alpha\beta = \frac{1}{\sigma_0^2 + \sigma_Z^2}$. Fig. 3 shows that a one-layer network of size $n = 16$ trained on tasks with $\sigma_Z^2 = 1, \sigma_0^2 = 2, d = 8, L = 500$ indeed achieves this bound, training to nearly diagonal weights with the appropriate scale $\langle w_{KQ}^{(ii)}\rangle\langle w_{PV}^{(ii)}\rangle = 0.327 \approx 1/3$ (similar weights are learned for each seed, up to a sign flip).

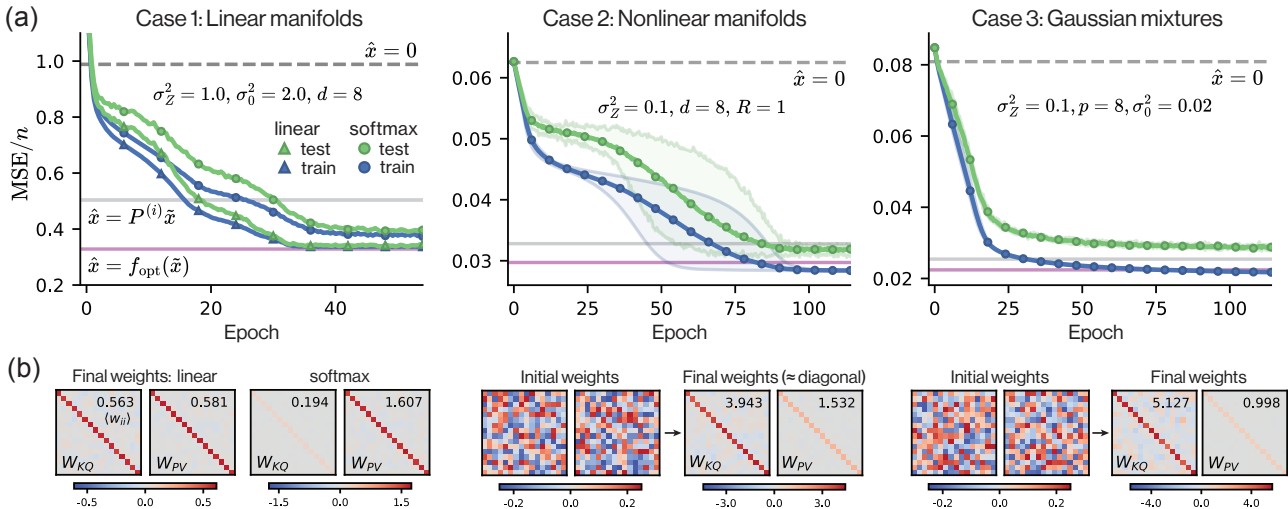

*Figure 3.* (a) Training dynamics for the studied cases using one-layer softmax attention (circles) as well as linear attention (triangles). Solid lines represent the average loss over six seeds, with the shaded area indicating the range for cases 2 and 3. For each case, the grey dashed baseline indicates the 0-predictor, and the pink line indicates the Bayes optimal predictor. All cases use a context length of $L = 500$, ambient dimension $n = 16$, and are trained with Adam on a dataset of size 800 with batch size 80 and standard weight initialization $w_{ij} \sim U[-1/\sqrt{n}, 1/\sqrt{n}]$. (b) Final attention weights $W_{KQ}$ and $W_{PV}$ are shown. For each, we indicate the mean of the diagonal elements. Representative initial weights are displayed for the second and third case.

Fig. 4(a) displays how this bound is approached as the context length $L$ of training samples is increased. In Fig. 4(b) we study how the performance of a model trained to denoise random subspaces of dimension $d = 8$ is affected by shifts in the subspace dimension at inference time. We find that when provided sufficient context, such models can adapt with mild performance loss to solve more challenging tasks not present in the training set.

It is evident from Fig. 3(a) that the softmax network performs similarly to the linear one for this task. We can understand this through the small argument expansion of the softmax function mentioned above. The learned weights displayed in Fig. 3(b) indicate that $\beta^{\text{softmax}} \approx 0.194$ becomes small (note it decreases by a factor $\epsilon \approx 0.344$ relative to $\beta^{\text{linear}}$), while the value scale $\alpha^{\text{softmax}} \approx 1.607$ becomes larger by a similar factor $\sim 1/\epsilon$ to compensate. Thus, although the optimal denoiser for this case is intuitively expressed through linear self-attention, it can also be achieved with softmax self-attention in the appropriate limit.

Moreover, we find that when the entire prompt undergoes a global invertible transformation $A \neq I$, the optimal attention weights are no longer scaled identity matrices but acquire a structured form determined by $A$. Both linear and softmax attention layers are able to recover this structure through training; see Appendix H for details and empirical verification.

### 3.3. Case 2 – Nonlinear manifolds

Fig. 3 (case 2) shows networks of size $n = 16$ trained to denoise subspheres of dimension $d = 8$ and radius $R = 1$, with corruption $\sigma_Z^2 = 0.1$ and context length $L = 500$. Once again, the network trains to have scaled identity weights.

We note that although the network nearly achieves the optimal MSE on the test set, the weights appear at first glance to deviate slightly from the Bayes optimal predictor of Section 2.3, which indicated $W_{PV} = \alpha I$, $W_{KQ} = \beta I$ with $\alpha = 1, \beta = 1/\sigma_Z^2$. To better understand this, we consider a coarse-grained MSE loss landscape by scanning over $\alpha$ and $\beta$. See Fig. 6(a) in Appendix G. We find that the 2D loss landscape has roughly hyperbolic level sets which is suggestive of the linear attention limit, where the weight scales become constrained by their product $\alpha\beta$. Reflecting the symmetry of the problem, we also note mirrored negative solutions (i.e. one could also identify $\alpha = -1, \beta = -1/\sigma_Z^2$ from the analysis in Section 2.3). Importantly, the plot shows that the trained network lies in the same valley of the loss landscape as the optimal predictor, in agreement with Fig. 3. Moreover, the shape of the loss landscape suggested that linear attention might also be applicable to this case, which we demonstrate and discuss further in Appendix G.

### 3.4. Case 3 – Gaussian mixtures

Figure 3 (case 3) shows networks of size $n = 16$ trained to denoise balanced Gaussian mixtures with $p = 8$ compo-

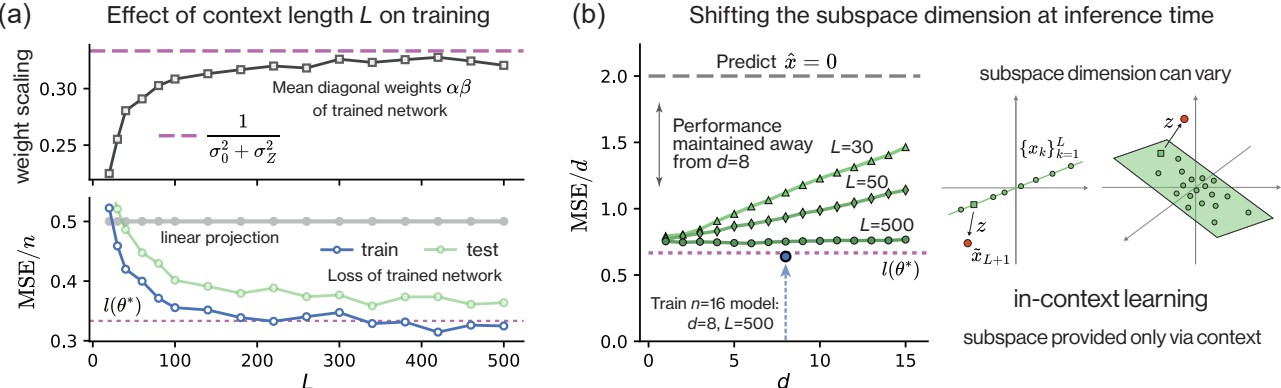

*Figure 4.* (a) Trained linear attention network converges to Bayes optimal estimator as context length increases ($n = 16$, $d = 8$, $\sigma_0^2 = 2, \sigma_z^2 = 1$). (b) A network trained to denoise subspaces of dimension $d = 8$ can accurately denoise subspaces of different dimensions presented at inference time, given sufficient context.

nents that have isotropic variance $\sigma_0^2 = 0.02$ and centers randomly placed on the unit sphere in $\mathbb{R}^n$. The corruption magnitude is $\sigma_Z^2 = 0.1$ and context length is $L = 500$. The baselines show the zero predictor (dashed grey line) as well as the optimum from Proposition (4) (pink) and its $\sigma_0^2 \to 0$ approximation Eq. (8) (grey).

The trained weights qualitatively approach the optimal estimator for the zero-variance limit but with a slightly different scaling: while the scale of $W_{PV}$ is $\alpha \approx 1$, the $W_{KQ}$ scale is $\beta \approx 5.127 < 1/\sigma_Z^2$. To study this, we provide a corresponding plot of the 2D loss landscape in Fig. 6(a) in Appendix G. While the symmetry of the previous case has been broken (the context cluster centers $\{\mu_a\}$ will not satisfy $\langle \mu \rangle = 0$), we again find that the trained network lies in the anticipated global valley of the MSE loss landscape.

## 4. Connection to dense associative memory networks

In each of the denoising problems studied above, we have shown analytically and empirically that the optimal weights of the one-layer transformer are scaled identity matrices $W_{PV} \approx \alpha I, W_{KQ} \approx \beta I$. In the softmax case, the trained denoiser can be concisely expressed as

$$\hat{x} = g(X_{1:L}, \tilde{x}) := \alpha X_{1:L} \text{softmax}(\beta X_{1:L}^T \tilde{x}),$$

re-written such that $X \in \mathbb{R}^{n \times L}$ stores pure context tokens.

We now demonstrate that such denoising corresponds to one-step gradient descent (with specific step sizes) of energy models related to dense associative memory networks, also known as modern Hopfield networks (Ramsauer et al., 2021; Demircigil et al., 2017; Krotov & Hopfield, 2016).

Consider the energy function:

$$\mathcal{E}(X_{1:L}, s) = \frac{1}{2\alpha} \|s\|^2 - \frac{1}{\beta} \log \left( \sum_{t=1}^{L} e^{\beta X_t^T s} \right), \quad (16)$$

which mirrors the Ramsauer et al. (2021) construction but with a Lagrange multiplier added to the first term. Figure 5 illustrates this energy landscape for the spherical manifold case.

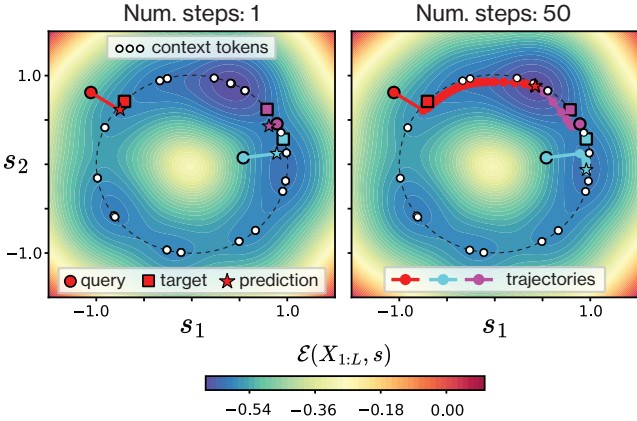

*Figure 5.* Gradient descent denoising for the nonlinear manifold case (spheres) in $n = 2$ with $d = 1$. A context-aware dense associative memory network $\mathcal{E}(X_{1:L}, s)$ is constructed whose gradient corresponds to the Bayes optimal update (trained attention layer). Note that the density of sampled context tokens sculpts the valleys of the energy landscape. Left: the attention step of a one-layer transformer trained on the denoising task corresponds to a single gradient descent step. Right: Iterating the denoising process—as is conventional for Hopfield networks—can potentially degrade the estimate by causing it to become query-independent (e.g. converging to a distant minimum). Here $R = 1, \sigma_Z^2 = 10, L = 20$ and $\alpha = 1, \beta = 1/\sigma_Z^2$.

An operation inherent to the associative memory perspective is the recurrent application of a denoising update. Gradient descent iteration $s(t + 1) = s(t) - \gamma \nabla_s \mathcal{E}(X_{1:L}, s(t))$ yields

$$s(t+1) = \left(1 - \frac{\gamma}{\alpha}\right) s(t) + \gamma X_{1:L}\text{softmax}\left(\beta X_{1:L}^T s(t)\right). \tag{17}$$

It is now clear that initializing the state to the query $s(0) = \tilde{x}$ and taking a single step with size $\gamma = \alpha$ recovers the behavior of the trained attention model (Fig. 5). The attention mechanism here is thus mechanistically interpretable: the context tokens $X_{1:L}$ induce a context-dependent associative memory landscape, while the query acts as an initial condition for inference-time gradient descent. One could naturally consider alternative step sizes and recurrent iteration. However, Fig. 5 demonstrates that naive iteration of Eq. (17) has the potential to degrade performance.

Additional details are provided in Appendix I. In particular, the energy model for linear attention is discussed in Appendix I.1.

## 5. Discussion

Motivated by the connection between attention mechanisms and dense associative memories, here we have introduced in-context denoising, a task that distills their relationship. We first analyze the general problem, deriving Bayes optimal predictors for certain restricted tasks. We identify that one-layer transformers using either softmax or linearized self-attention are expressive enough to describe these predictors. We then empirically demonstrate that standard training of attention layers from random initial weights will readily converge to scaled identity weights with scales that approach the derived optima given sufficient context. Accordingly, the rather minimal transformers studied here can perform optimal denoising of novel tasks provided at inference time via self-contained prompts. This work therefore sheds light on other in-context learning phenomena, a point we return to below.

While practical transformers differ in various ways from the minimal models studied here, we note several key connections. Intriguingly, the self-attention heads of trained transformers sometimes exhibit weights $W_{KQ}, W_{PV}$ that resemble scaled identity matrices, i.e. $cI + \epsilon$ with small fluctuations $\epsilon_{ij} \sim \mathcal{N}(0, \sigma^2)$, an observation noted in Trockman & Kolter (2023). This phenomenon motivated their proposal of "mimetic" weight initialization schemes mirroring this learned structure. Relatedly, connections to associative memory concepts have been explored in other architectures (Smart & Zilman, 2021), which enabled data-dependent weight initialization strategies to be identified and leveraged.

More broadly, our study suggests that trained attention layers can readily adopt structures that facilitate context-aware associative retrieval. We have also noted preliminary connections between our work and other architectural features of modern transformers, namely layer normalization and residual streams, which warrant further study.

In-context denoising and generative modeling both involve learning about an underlying distribution, suggesting potential relationships between these two tasks. Recently, Pham et al. (2024) invoked spurious states of the Hopfield model as a way of understanding how one can move away from retrieving individual memorized patterns towards generalization via appropriate mixtures of multiple similar "memories". In our work, one-step updates do not have to land in a spurious minimum, but we often operate under circumstances where there are such states (see, for example, the energy landscape in Fig. 5). More generally, analogies between energy-based associative memory and diffusion models have recently been noted (Ambrogioni, 2024; Hoover et al., 2024b). Lastly, Bayes optimal denoisers play an important role in the analysis (Ghio et al., 2024) of a very related generative model that is based on stochastic interpolants (Albergo & Vanden-Eijnden, 2023). Although this work focuses on the case where it is possible to sample enough tokens from the relevant distributions for certain functions to converge, generative models become important when the distribution is in a prohibitively high-dimensional space making direct sampling difficult. Nonetheless, investigating the precise relationship between our work and different generative modeling approaches would be an interesting direction to pursue.

Overall, this work refines the connection between dense associative memories and attention layers first identified in (Ramsauer et al., 2021). While we show that one energy minimization step of a particular DAM (associated with a trained attention layer) is optimal for the denoising tasks studied here, it remains an open question whether multilayer architectures with varying or tied weights could extend these results to more complex tasks by effectively performing multiple iterative steps. This aligns with recent studies on in-context learning, which have considered whether transformers with multiple layers emulate gradient descent updates on a context-specific objective (Von Oswald et al., 2023; Shen et al., 2024; Dai et al., 2023; Ahn et al., 2023), and may provide a bridge to work on emerging architectures guided by associative memory principles (Hoover et al., 2023). Investigating when and how multilayer attention architectures perform such gradient descent iterations in a manner that is both context-dependent and informed by a large training set represents an exciting direction for future research at the intersection of transformer mechanisms, associative memory retrieval, and in-context learning.

## Software and Data

Python code underlying this work is available at https://github.com/mattsmart/in-context-denoising.

## Acknowledgements

MS acknowledges M. Mézard for very useful feedback on an earlier version of this work. AS thanks D. Krotov and P. Mehta for enlightening discussions on related matters. Our early work also benefited from AS's participation in the deeplearning23 workshop at the Kavli Institute for Theoretical Physics (KITP), which was supported in part by grants NSF PHY-1748958 and PHY-2309135 to KITP. AS thanks Y. Bahri and C. Pehlevan for their patience and willingness to listen to our early ideas at KITP.

## Impact Statement

This paper presents work whose goal is to advance the field of Machine Learning. There are many potential societal consequences of our work, none which we feel must be specifically highlighted here.

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

## A. Notation

### A.1. Recurring notation

- $n$ – ambient dimension of input tokens.

- $x_t \in \mathbb{R}^n$ – the value of the $t$-th random input token.

- $E = (X_1, ..., X_L, \tilde{X})$ – the random variable input to the sequence model. The "tilde" indicates that the final token has in some way been corrupted. $E$ takes values $(x_1, ..., x_L, \tilde{x}) \in \mathbb{R}^{n \times (L+1)}$. Note: while capital $X$ or $X_i$ here denotes a random variable, in Section D use $X_{1:L}$ or simply $X$ to refer to the realized matrix of input tokens.

- $L$ – context length = number of uncorrupted tokens.

- $d$ – dimensionality of manifold $S$ that $x_t$ are sampled from

- $N$ – number of training pairs

### A.2. Bayes posterior notation

- $p_X(x)$ is task-dependent (the three scenarios considered here are introduced above).

- $p_{\tilde{X}}(\tilde{x})$ where $\tilde{x} = x + z$. For a sum of independent random variables, $Y = X_1 + X_2$, their pdf is a convolution $p_Y(y) = \int p_{X_1}(x) p_{X_2}(y - x) dx$. Thus:

$$p_{\tilde{X}}(\tilde{x}) = \int p_Z(z) p_X(\tilde{x} - z) dz$$
$$= C_Z \int e^{-\|z\|^2 / 2\sigma_Z^2} p_X(\tilde{x} - z) dz$$

where $C_Z = (2\pi\sigma_Z^2)^{-n/2}$ is a constant.

- $p_{\tilde{X}|X}(\tilde{x} \mid x)$: This is simply

$$p_Z(\tilde{x} - x) = C_Z e^{-\|\tilde{x} - x\|^2 / 2\sigma_Z^2}.$$

- $p_{X|\tilde{X}}(x \mid \tilde{x})$: By Bayes' theorem, this is

$$p_{X|\tilde{X}}(x \mid \tilde{x}) = \frac{p_{\tilde{X}|X}(\tilde{x} \mid x) p_X(x)}{p_{\tilde{X}}(\tilde{x})}$$
$$= \frac{e^{-\|\tilde{x} - x\|^2 / 2\sigma_Z^2} p_X(x)}{\int e^{-\|\tilde{x} - x'\|^2 / 2\sigma_Z^2} p_X(x') dx'}.$$

- Posterior mean:

$$\mathbb{E}_{X|\tilde{X}}[X \mid \tilde{X}] = \int x \, p_{X|\tilde{X}}(x \mid \tilde{X}) dx$$
$$= \frac{\int x \, e^{-\|\tilde{X} - x\|^2 / 2\sigma_Z^2} p_X(x) dx}{\int e^{-\|\tilde{X} - x\|^2 / 2\sigma_Z^2} p_X(x) dx}.$$

# B. Bayes optimal predictors for square loss

## B.1. Proof of Proposition 1

*Proof.* Observe that

$$
\begin{aligned}
\mathbb{E}\left[\|X - f(\tilde{X})\|^2\right] &= \mathbb{E}_{\tilde{X}}\left[\mathbb{E}_{X|\tilde{X}}\left[\|X - f(\tilde{X})\|^2 \mid \tilde{X}\right]\right] \\
&= \mathbb{E}_{\tilde{X}}\left[\mathbb{E}_{X|\tilde{X}}\left[\|X - \mathbb{E}[X \mid \tilde{X}]\|^2 \mid \tilde{X}\right] \right. \\
&\qquad \left. + \|\mathbb{E}[X \mid \tilde{X}] - f(\tilde{X})\|^2\right] \\
&\geq \mathbb{E}_{\tilde{X}}\left[\mathbb{E}_{X|\tilde{X}}\left[\|X - \mathbb{E}[X \mid \tilde{X}]\|^2 \mid \tilde{X}\right]\right] \\
&= \mathbb{E}_{\tilde{X}}\left[\operatorname{Tr}\operatorname{Cov}(X \mid \tilde{X})\right].
\end{aligned}
$$

Note the final line is independent of $f$. This inequality becomes an equality when $f(\tilde{X}) = \mathbb{E}[X \mid \tilde{X}]$. $\qquad\square$

# C. Details of Bayes optimal denoising baselines for each case

## C.1. Proof of Proposition 2

*Proof.* The linear denoising task is a special case of the result in Proposition 1. Here, $X$ is an isotropic Gaussian in a restricted subspace,

$$
p_{X|\tilde{X}}(x \mid \tilde{x}) = C(\tilde{x})p_X(x)e^{-\frac{\|x-\tilde{x}\|^2}{2\sigma_Z^2}}
$$

where $C(\tilde{x})$ is a normalizing factor. The noise can be decomposed into parallel and perpendicular parts using the projection $P$ onto $S$, i.e.

$$
\tilde{X} = \tilde{X}_\| + \tilde{X}_\perp = P\tilde{X} + (I - P)\tilde{X},
$$

so that

$$
e^{-\frac{\|x-\tilde{x}\|^2}{2\sigma_Z^2}} = e^{-\frac{\|x-\tilde{x}_\|\|^2}{2\sigma_Z^2}}\, e^{-\frac{\|\tilde{x}_\perp\|^2}{2\sigma_Z^2}}.
$$

Only the first factor matters for $p_{X|\tilde{X}}(x \mid \tilde{x})$ since it depends on $x$. Then, for $x \in S$, the linear subspace supporting $p_X$, dropping the $x$ independent $\tilde{x}_\perp$ contribution,

$$
\begin{aligned}
p_X(x)e^{-\frac{\|x-\tilde{x}_\|\|^2}{2\sigma_Z^2}} &\propto e^{-\frac{\|x\|^2}{2\sigma_0^2} - \frac{\|x-\tilde{x}_\|\|^2}{2\sigma_Z^2}} \\
&\propto \exp\left(-\frac{\|x - \frac{\sigma_0^2}{\sigma_0^2+\sigma_Z^2}\tilde{x}_\|\|^2}{2\frac{\sigma_0^2\sigma_Z^2}{\sigma_0^2+\sigma_Z^2}}\right).
\end{aligned}
$$

Thus, $f(\tilde{X}) = \frac{\sigma_0^2}{\sigma_0^2+\sigma_Z^2}\tilde{X}_\| = \frac{\sigma_0^2}{\sigma_0^2+\sigma_Z^2}P\tilde{X}$.

Using $\tilde{X} = X + Z$, $X = PX$, and the independence of $X$ and $Z$

$$
\mathbb{E}\left[\|X - \frac{\sigma_0^2}{\sigma_0^2+\sigma_Z^2}P\tilde{X}\|^2\right] = \mathbb{E}\left[\|\frac{\sigma_Z^2}{\sigma_0^2+\sigma_Z^2}PX\|^2\right] + \mathbb{E}\left[\|\frac{\sigma_0^2}{\sigma_0^2+\sigma_Z^2}PZ\|^2\right] = \frac{\sigma_Z^4 d\sigma_0^2 + \sigma_0^4 d\sigma_Z^2}{(\sigma_0^2+\sigma_Z^2)^2} = \frac{d\sigma_0^2\sigma_Z^2}{\sigma_0^2+\sigma_Z^2}.
$$

$\qquad\square$

## C.2. Proof of Proposition 3

*Proof.* In the nonlinear manifold denoising problem, we focus on the case of lower dimensional spheres $S$ (e.g. the circle $S^1 \subset \mathbb{R}^2$). For such manifolds, we have

$$
\mathbb{E}[X \mid \tilde{X} = \tilde{x}] = \frac{\int e^{-\frac{\|x - \tilde{x}_\|\|^2}{2\sigma_Z^2}} \, x \, p_X(x) dx}{\int e^{-\frac{\|x - \tilde{x}_\|\|^2}{2\sigma_Z^2}} \, p_X(x) dx}
$$

$$
= \frac{\int e^{\langle x, \tilde{x}_\| \rangle / \sigma_Z^2} \, x \, dS_x}{\int e^{\langle x, \tilde{x}_\| \rangle / \sigma_Z^2} \, dS_x}.
$$

We have used the fact that $\|x - \tilde{x}_\|\|^2 = \|x\|^2 + \|\tilde{x}_\|\|^2 - 2\langle x, \tilde{x}_\| \rangle$ and that $\|x\|$ is fixed on the sphere.

The integrals can be evaluated directly once the parameters are specified. If $S$ is a $d$–sphere of radius $R$, then the optimal predictor is again a shrunk projection of $\tilde{x}$ onto $S$,

$$
\frac{\int_0^\pi e^{R\|\tilde{x}_\|\| \cos\theta / \sigma_Z^2} \, \cos\theta \, \sin^{(d-1)}\theta \, d\theta}{\int_0^\pi e^{R\|\tilde{x}_\|\| \cos\theta / \sigma_Z^2} \, \sin^{(d-1)}\theta \, d\theta} R \frac{\tilde{x}_\|}{\|\tilde{x}_\|\|}
$$

$$
= \frac{I_{\frac{d+1}{2}}\left(R\frac{\|\tilde{x}_\|\|}{\sigma_Z^2}\right)}{I_{\frac{d-1}{2}}\left(R\frac{\|\tilde{x}_\|\|}{\sigma_Z^2}\right)} R \frac{\tilde{x}_\|}{\|\tilde{x}_\|\|},
$$

where we used identities involving $I_\nu(y)$, modified Bessel function of the first kind of order $\nu$ (Gradshteyn & Ryzhik, 2007). The vector $R\frac{\tilde{x}_\|}{\|\tilde{x}_\|\|}$ is the point on $S$ in the direction of $x_\|$.

$\square$

## C.3. Proof of Proposition 4

*Proof.* For the clustering case involving isotropic Gaussian mixtures with parameters $\{w_a, (\mu_a, \sigma_a^2)\}_{a=1}^p$,

$$
\mathbb{E}[X \mid \tilde{X} = \tilde{x}] = \frac{\int e^{-\frac{\|x - \tilde{x}\|^2}{2\sigma_Z^2}} \sum_a \left(w_a C_a e^{-\frac{\|x - \mu_\alpha\|^2}{2\sigma_a^2}}\right) x \, dx}{\int e^{-\frac{\|x - \tilde{x}\|^2}{2\sigma_Z^2}} \sum_a \left(w_a C_a e^{-\frac{\|x - \mu_a\|^2}{2\sigma_a^2}}\right) dx},
$$

where $C_a = (2\pi\sigma_a^2)^{-\frac{n}{2}}$.

We can simplify this expression by completing the square in the exponent and using the fact that the integral of a Gaussian about its mean is zero. This yields

$$
\mathbb{E}[X \mid \tilde{X} = \tilde{x}] = \frac{\sum_a w_a C_a m_a \int \exp(-g_a) \, dx}{\sum_a w_a C_a \int \exp(-g_a) \, dx}
$$

where we have introduced

$$
g_a = \frac{1}{2}\left(\frac{\sigma_Z^2 + \sigma_a^2}{\sigma_Z^2 \sigma_a^2}\right) \|x - m_\alpha\|^2 + \frac{1}{2(\sigma_Z^2 + \sigma_a^2)} \|\tilde{x} - \mu_a\|^2,
$$

with

$$
m_a = \frac{\sigma_a^2 \tilde{x} + \sigma_Z^2 \mu_a}{\sigma_a^2 + \sigma_Z^2}.
$$

Doing the integrals and using the expressions for $C_a, m_a$

$$\mathbb{E}[X \mid \tilde{X} = \tilde{x}] = \frac{\sum_a w_a \left(\frac{\sigma_Z^2 + \sigma_a^2}{\sigma_a^2}\right)^{n/2} \exp\left(-\frac{\|\tilde{x} - \mu_a\|^2}{2(\sigma_Z^2 + \sigma_a^2)}\right)\left(\frac{\sigma_a^2 \tilde{x} + \sigma_Z^2 \mu_a}{\sigma_a^2 + \sigma_Z^2}\right)}{\sum_a w_a \left(\frac{\sigma_Z^2 + \sigma_a^2}{\sigma_a^2}\right)^{n/2} \exp\left(-\frac{\|\tilde{x} - \mu_a\|^2}{2(\sigma_Z^2 + \sigma_a^2)}\right)}$$

In the case that the center norms $\|\mu_a\|$ are independent of $a$ and variances $\sigma_a^2 = \sigma_0$, we have

$$\mathbb{E}[X \mid \tilde{X} = \tilde{x}] = \frac{\sigma_0^2}{\sigma_0^2 + \sigma_Z^2}\, \tilde{x} + \frac{\sigma_Z^2}{\sigma_0^2 + \sigma_Z^2} \frac{\sum_a w_a \mu_a \exp\left(\frac{\langle \tilde{x}, \mu_a \rangle}{\sigma_Z^2 + \sigma_0^2}\right)}{\sum_a w_a \exp\left(\frac{\langle \tilde{x}, \mu_a \rangle}{\sigma_Z^2 + \sigma_0^2}\right)}.$$

Note that in the limit that $\sigma_0 \to 0$, this becomes expressible by one-layer self-attention, since one can simply replace the matrix of cluster centers $M = [\mu_1 \dots \mu_p]$ implicit in the expression with the context $X_{1:L}$ itself,

$$\mathbb{E}[X \mid \tilde{X}] = \frac{\sum_a w_a e^{\langle \mu_\alpha, \tilde{X} \rangle / \sigma_Z^2} \mu_a}{\sum_a w_a e^{\langle \mu_\alpha, \tilde{X} \rangle / \sigma_Z^2}}.$$

$\square$

## D. Additional details on attention layers and softmax expansion

### D.1. Standard self-attention

Given a sequence of $L_{\text{seq}}$ input tokens $x_i \in \mathbb{R}^n$ represented as a matrix $X \in \mathbb{R}^{n \times L_{\text{seq}}}$, standard self-attention defines query, key, and value matrices

$$K = W_K X, Q = W_Q X, V = W_V X \tag{A.1}$$

where $W_K, W_Q \in \mathbb{R}^{n_{\text{attn}} \times n}$ and $W_V \in \mathbb{R}^{n_{\text{out}} \times n}$. The softmax self-attention map (Vaswani et al., 2017) is then

$$\text{Attn}(X, W_V, W_K^T W_Q) := V \text{softmax}(K^T Q) \in \mathbb{R}^{n_{\text{out}} \times L_{\text{seq}}}. \tag{A.2}$$

On merging $W_K, W_Q$ into $W_{KQ} = W_K^T W_Q$: The simplification $W_{KQ} = W_K^T W_Q$ (made here and elsewhere) is general only when $n_{\text{attn}} \geq n$; in that case, the product $W_{KQ}$ can have rank $n$ and thus it is reasonable to work with the combined matrix. On the other hand, if $n_{\text{attn}} < n$, then the rank of their product is at most $n_{\text{attn}}$ and thus there are matrices in $\mathbb{R}^{n \times n}$ that cannot be expressed as $W_K^T W_Q$. A similar point can be made about $W_{PV}$. We note that while $n_{\text{attn}} < n$ may be used in practical settings, one often also uses multiple heads which when concatenated could be (roughly) viewed as a single higher-rank head.

We will also use the simplest version of linear attention (Katharopoulos et al., 2020),

$$\text{Attn}_{\text{Lin}}(X, W_V, W_K^T W_Q) := \frac{1}{L_{\text{seq}}} V(K^T Q) \in \mathbb{R}^{n_{\text{out}} \times L_{\text{seq}}}. \tag{A.3}$$

### D.2. Minimal transformer architecture for denoising

We now consider a simplified one-layer linear transformer in term of our variable $E = (X_{1:L}, \tilde{X})$ taking values in $\mathbb{R}^{n \times (L+1)}$ and start with the linear transformer which still has sufficient expressive power to capture our finite sample approximation to the Bayes optimal answer in the linear case. Inspired by Zhang et al. (2024), we define

$$\text{Attn}_{\text{Lin}}(E, W_{PV}, W_{KQ}) := \frac{1}{L} W_{PV} E M_{\text{Lin}} E^T W_{KQ} E \tag{A.4}$$

taking values in $\mathbb{R}^{n \times (L+1)}$. The additional aspect compared to the last subsection is the masking matrix $M_{\text{Lin}} \in \mathbb{R}^{(L+1) \times (L+1)}$ which is of the form

$$M_{\text{Lin}} = \begin{bmatrix} I_L & 0_{L \times 1} \\ 0_{1 \times L} & 0 \end{bmatrix}, \tag{A.5}$$

preventing $W_{PV}\tilde{X}$ from being added to the output.

Note that this more detailed expression is equivalent to the form used in the main text.

$$\hat{X} = F_{\text{Lin}}(E, \theta) := \frac{1}{L} W_{PV} X_{1:L} X_{1:L}^T W_{KQ} \tilde{X}$$

With learnable weights $W_{KQ}, W_{PV} \in \mathbb{R}^{n \times n}$ abbreviated by $\theta$, we define

$$F(E, \theta) := [\text{Attn}_{\text{Lin}}(E, W_{PV}, W_{KQ})]_{:,L+1}. \tag{A.6}$$

Note that, when $W_{PV} = \alpha I_n, W_{KQ} = \beta I_n$, and $\alpha\beta = \frac{1}{\sigma_0^2 + \sigma_Z^2}$, $F(E, \theta)$ should approximate the Bayes optimal answer $f_{\text{opt}}(\tilde{X})$ as $L \to \infty$.

Similarly, we could argue that the second two problems, the $d$-dimesional spheres and the $\sigma_0 \to 0$ zero limit of the Gaussian mixtures could be addressed by the full softmax attention

$$\text{Attn}(E, W_{PV}, W_{KQ}) = W_{PV} E \text{softmax}(E^T W_{KQ} E + M) \tag{A.7}$$

taking values in $\mathbb{R}^{n \times (L+1)}$ where $M \in \bar{\mathbb{R}}^{(L+1) \times (L+1)}$ is a masking matrix of the form

$$M = \begin{bmatrix} 0_{L \times (L+1)} \\ (-\infty)1_{1 \times L+1} \end{bmatrix}, \tag{A.8}$$

once more, preventing the contribution of $\tilde{X}$ value to the output. The function $\text{softmax}(z) := \frac{1}{\sum_{i=1}^n e^{z_i}}(e^{z_1}, \dots, e^{z_n})^T \in \mathbb{R}^n$ is applied column-wise.

We then define

$$F(E, \theta) := [\text{Attn}(E, W_{PV}, W_{KQ})]_{:,L+1}, \tag{A.9}$$

which is equivalent to the simplified form used in the main text:

$$\hat{X} = F(E, \theta) := W_{PV} X_{1:L} \text{softmax}(X_{1:L}^T W_{KQ} \tilde{X}).$$

### D.3. Proof of Theorem 3.1

*Proof.* Let the support of $p_X$ be a subset of a sphere, centered around the origin, of radius $R$. Then the function

$$g(\{X_t\}_{t=1}^L, \tilde{x}) = \frac{\sum_{t=1}^L X_t e^{\langle X_t, \tilde{x} \rangle / \sigma_Z^2}}{\sum_{t=1}^L e^{\langle X_t, \tilde{x} \rangle / \sigma_Z^2}} = \frac{\frac{1}{L}\sum_{t=1}^L X_t e^{\langle X_t, \tilde{x} \rangle / \sigma_Z^2}}{\frac{1}{L}\sum_{t=1}^L e^{\langle X_t, \tilde{x} \rangle / \sigma_Z^2}}. \tag{A.10}$$

Both the numerator $\frac{1}{L}\sum_{t=1}^L X_t e^{\langle X_t, \tilde{x} \rangle / \sigma_Z^2}$ and the denominator $\frac{1}{L}\sum_{t=1}^L e^{\langle X_t, \tilde{x} \rangle / \sigma_Z^2}$ are averages of independent and identically distributed bounded random variables. By the strong law of large numbers, as $L \to \infty$, the average vector in the numerator converges to almost surely to $\int e^{\langle x, \tilde{x}_\parallel \rangle / \sigma_Z^2} x \, dp_X(x)$ for each component, while the average in the denominator almost surely converges $\int e^{\langle x, \tilde{x}_\parallel \rangle / \sigma_Z^2} dp_X(x)$, which is positive. So, as $L \to \infty$, the ratio in Eq. A.10 converges almost surely to

$$\frac{\int e^{\langle x, \tilde{x}_\parallel \rangle / \sigma_Z^2} x \, dp_X(x)}{\int e^{\langle x, \tilde{x}_\parallel \rangle / \sigma_Z^2} dp_X(x)},$$

which is the Bayes optimal answer $f_{\text{opt}}(\tilde{x})$ for all $\tilde{x} \in \mathbb{R}^n$. $\square$

## E. Further discussion of convergence rates as $L \to \infty$ and the dependence on dimensions

Our analysis primarily focused on the asymptotic behavior as $L \to \infty$ using the strong law of large numbers, which just requires the mean to exist (Loève, 1977). However, in the linear example, our tokens are Gaussian, and in the two nonlinear cases they are bounded. Intuitively, we expect error $O(\frac{1}{\sqrt{L}})$. In fact, we can give precise results of the form that the probability of the difference between the empirical sum for the ideal weights departing from the expectation by less

than $C(\tilde{x})\sqrt{\frac{f\left(d,\ln\frac{1}{\delta}\right)}{L}}$ is greater than $1 - \delta$. The function $C$ of the query vector and the function $f$ depend on the problem. Interestingly, these bounds depend on $d$, the dimension spanned by the tokens, not the ambient dimension $n$.

As mentioned before, the results of the previous paragraph refer to the convergence of the finite sample attention expressions for ideal weights, namely those corresponding to Bayes optimal answer. There is a second source of error associated with finite sample estimation of weights, which should also get small as $L$ becomes large. Once more the expectation is that the weights are known to error $O(\frac{1}{\sqrt{L}})$ for well-converged training procedures, although this is more difficult to guarantee or quantify analytically. Overall we expect the loss (MSE) to go down inversely with some power of $L$. Fig. 4(a) provides some empirical evidence for this relationship, showing how performance improves with increasing context length.

Notice that the one-layer transformer output is a linear combination of the uncorrupted samples. Hence, if the distribution $p_X$ is supported by a $d$-dimensional linear subspace, the estimate $\hat{X}$ is also in that subspace. We can therefore look at convergence restricted to the supporting subspace. Therefore, it is the dimensionality of the supporting subspace that matters.

Let a $d$-dimensional vector space $V$ be a linear subspace of $\mathbb{R}^n$. We define the maximum norm for $V$ with respect to some orthonormal basis $\{v_i\}_{i=1}^d$ in $V$ as $||x||_{\infty,V} := \max_{i \in \{1,...,d\}} |\langle v_i, x \rangle|$ for any $x \in V$. The conventional maximum norm for $\mathbb{R}^n$, of course, is defined with respect to the standard orthonormal basis $\{e_j\}_{j=1}^n$. Since $|\langle v_i, x \rangle| \leq ||x||_{\infty,V}$, for all $i$,

$$||x||_2^2 = \sum_{i=1}^d (\langle v_i, x \rangle)^2 \leq d||x||_{\infty,V}^2 \implies ||x||_2 \leq \sqrt{d}||x||_{\infty,V}.$$

Then, for any $x \in V \subseteq \mathbb{R}^n$, $||x||_\infty \leq \sqrt{d}||x||_{\infty,V}$, since $|\langle x, e_j \rangle| \leq ||x||_2 \leq \sqrt{d}||x||_{\infty,V}$, for all $j \in \{1 \ldots, n\}$. Thus, controlling component-wise error in any orthonormal basis in $V$ controls component-wise error in $\mathbb{R}^n$, in an $n$-independent but $d$-dependent manner. In the following, we give a flavor of how we can analyze finite sample estimate errors in $V$. The maximum norm $|| \cdot ||_\infty$ is to be understood as $|| \cdot ||_{\infty,V}$ for some orthonormal basis choice. Here is the result relevant to the linear case described Subsection 2.2.1.

**Proposition 5.** *Let* $X_t \overset{i.i.d}{\sim} \mathcal{N}(0, \sigma_0^2 I_d), t = 1, \ldots, L$ *and let* $\hat{\Pi} := \frac{1}{\sigma_0^2 L}\sum_{t=1}^L X_t X_t^T$. *Then, for any* $\delta \in (0,1)$,

$$Pr\left[||\hat{\Pi}\tilde{x} - \tilde{x}||_\infty < C||\tilde{x}||_2 \max\left\{\sqrt{\frac{d + \ln(\frac{2}{\delta})}{L}}, \frac{d + \ln(\frac{2}{\delta})}{L}\right\}\right] > 1 - \delta$$

*for some* $C > 0$.

*Proof.* We start by bounding the maximum norm of the difference,

$$||\hat{\Pi}\tilde{x} - \tilde{x}||_\infty \leq ||\hat{\Pi}\tilde{x} - \tilde{x}||_2 \leq ||\hat{\Pi} - I_d||_{\text{op}}||x||_2,$$

where $|| \cdot ||_{\text{op}}$ is the operator norm.

It can be shown that, for any $\delta \in (0,1)$

$$\Pr\left[||\hat{\Pi} - I_d||_{\text{op}} < C \max\left\{\sqrt{\frac{d + \ln(\frac{2}{\delta})}{L}}, \frac{d + \ln(\frac{2}{\delta})}{L}\right\}\right] > 1 - \delta$$

for some $C > 0$ (Rigollet & Hütter, 2023). Combining with the first bound, we get our result. $\square$

As to the nonlinear cases, the key result of Theorem 3.1 is the convergence of the numerator $\frac{1}{L}\sum_{t=1}^L X_t e^{\langle X_t, \tilde{x}_\parallel \rangle / \sigma_Z^2}$ to $\mathbb{E}[X e^{\langle X, \tilde{x}_\parallel \rangle / \sigma_Z^2}] = \int e^{\langle x, \tilde{x}_\parallel \rangle / \sigma_Z^2} x \, dp_X(x)$ and the denominator $\frac{1}{L}\sum_{t=1}^L e^{\langle X_t, \tilde{x}_\parallel \rangle / \sigma_Z^2}$ to $\mathbb{E}[e^{\langle X, \tilde{x}_\parallel \rangle / \sigma_Z^2}] = \int e^{\langle x, \tilde{x}_\parallel \rangle / \sigma_Z^2} dp_X(x)$.

In the following, we assume that the support of $p_X$ is inside a vector space $V$ whose dimension we denote by $d$ (instead of $d + 1$, as in the sphere problem). In addition, we refer to the projection of the query on $V$ by $\tilde{x} \in V$, instead of $\tilde{x}_\parallel$. As usual, the maximum norm in $V$ is with respect to some orthonormal basis choice

**Proposition 6.** *Let $X_t \overset{i.i.d}{\sim} p_X$ and $||X_t||_2 \leq R$ for $t = 1, \ldots, L$.*

*Then, for any $\delta \in (0, 1)$,*

$$Pr\left[\left|\frac{1}{L}\sum_{t=1}^{L} e^{\langle X_t, \tilde{x}\rangle/\sigma_Z^2} - \mathbb{E}[e^{\langle X, \tilde{x}\rangle/\sigma_Z^2}]\right| < \sinh\left(\frac{R||\tilde{x}||_2}{\sigma_Z^2}\right)\sqrt{\frac{2}{L}\ln\left(\frac{2}{\delta}\right)}\right] \geq 1 - \delta$$

*and*

$$Pr\left[\left|\left|\frac{1}{L}\sum_{t=1}^{L} X_t e^{\langle X_t, \tilde{x}\rangle/\sigma_Z^2} - \mathbb{E}[X e^{\langle X, \tilde{x}\rangle/\sigma_Z^2}]\right|\right|_{\infty} < R e^{\frac{R||\tilde{x}||_2}{\sigma_Z^2}}\sqrt{\frac{2}{L}\ln\left(\frac{2d}{\delta}\right)}\right] \geq 1 - \delta.$$

*Proof.* We provide the sketch of our proof here, the key ingredient of which is the Hoeffding inequality (Hoeffding, 1994).

For the average $\frac{1}{L}\sum_{t=1}^{L} e^{\langle X_t, \tilde{x}\rangle/\sigma_Z^2}$, each term in the sum is bounded above and below by $e^{\pm\frac{R||\tilde{x}||_2}{\sigma_Z^2}}$. So, the Hoeffding inequality leads to

$$\Pr\left[\left|\frac{1}{L}\sum_{t=1}^{L} e^{\langle X_t, \tilde{x}\rangle/\sigma_Z^2} - \mathbb{E}[e^{\langle X, \tilde{x}\rangle/\sigma_Z^2}]\right| \geq \epsilon\right] \leq 2\exp\left[-\frac{2L\epsilon^2}{\left(\exp\left(\frac{R||\tilde{x}||_2}{\sigma_Z^2}\right) - \exp\left(-\frac{R||\tilde{x}||_2}{\sigma_Z^2}\right)\right)^2}\right] = 2\exp\left[-\frac{L\epsilon^2}{2\sinh^2\left(\frac{R||\tilde{x}||_2}{\sigma_Z^2}\right)}\right].$$

Setting $\delta = 2\exp\left[-\frac{L\epsilon^2}{2\sinh^2\left(\frac{R||\tilde{x}||_2}{\sigma_Z^2}\right)}\right]$, we get $\epsilon = \sinh\left(\frac{R||\tilde{x}||_2}{\sigma_Z^2}\right)\sqrt{\frac{2}{L}\ln\left(\frac{2}{\delta}\right)}$, which gives our first probabilistic inequality.

For each component of the vector average $\frac{1}{L}\sum_{t=1}^{L} X_t e^{\langle X_t, \tilde{x}\rangle/\sigma_Z^2}$, the terms in the sum are bounded above and below by $\pm R e^{\frac{R||\tilde{x}||_2}{\sigma_Z^2}}$. We use similar arguments involving the Hoeffding inequality, combined with the union bound over all $d$ coordinates

$$\Pr\left[\left|\left|\frac{1}{L}\sum_{t=1}^{L} X_t e^{\langle X_t, \tilde{x}\rangle/\sigma_Z^2} - \mathbb{E}[X e^{\langle X, \tilde{x}\rangle/\sigma_Z^2}]\right|\right|_{\infty} \geq \epsilon\right] \leq 2d\exp\left[-\frac{L\epsilon^2}{2R^2\exp\left(\frac{2R||\tilde{x}||_2}{\sigma_Z^2}\right)}\right].$$

Once more, setting the RHS to $\delta$ and solving for $\epsilon$, we get our second probabilistic inequality. $\square$

## F. Limiting behaviors of the softmax function and softmax attention

**For small argument**

A Taylor expansion of the softmax function at zero gives

$$\text{softmax}(\beta v) = \frac{1}{Z}\left(\mathbb{1}_L + \beta v + O(\beta^2)\right),$$

where $Z = \sum_i \left(1 + \beta v_i + O(\beta^2)\right) = L(1 + \beta\bar{v} + O(\beta^2))$ is a normalizing factor, with $\bar{v} = \frac{1}{L}\sum_i v_i$. The notation $\mathbb{1}_L$ stands for an $L$-dimensional vector of ones.

Thus, we have

**Lemma F.1** (Small argument expansion of softmax). *As $\beta \to 0$,*

$$\text{softmax}(\beta v) = \frac{1}{L(1 + \beta\bar{v} + O(\beta^2))}\left(\mathbb{1}_L + \beta v + O(\beta^2)\right) = \frac{1}{L}\left(\mathbb{1}_L + \beta(v - \bar{v}\mathbb{1}) + O(\beta^2)\right).$$

### F.1. Proof of Proposition 3.2

*Proof.*

$$F\left(E, \left(\frac{1}{\epsilon}W_{PV}, \epsilon W_{KQ}\right)\right) := \frac{1}{\epsilon}W_{PV}X_{1:L}\text{softmax}(\epsilon X_{1:L}^T W_{KQ}\tilde{X}).$$

Using Lemma F.1, as $\epsilon \to 0$,

$$F\left(E, \left(\frac{1}{\epsilon}W_{PV}, \epsilon W_{KQ}\right)\right) = \frac{1}{\epsilon}W_{PV}X_{1:L}\left[\frac{1}{L}\left(\mathbb{1}_L + \epsilon\left(X_{1:L}^T W_{KQ}\tilde{X} - \left(\frac{1}{L}\sum_t X_t^T W_{KQ}\tilde{X}\right)\mathbb{1}_L\right) + O(\epsilon^2)\right)\right]$$

$$= \frac{1}{\epsilon}W_{PV}\bar{X} + \frac{1}{L}W_{PV}\sum_{t=1}^L X_t(X_t - \bar{X})^T W_{KQ}\tilde{X} + O(\epsilon), \tag{A.11}$$

where $\bar{X} = \frac{1}{L}\sum_{t=1}^L X_t$ is the empirical mean and the notation $\mathbb{1}_L$ emphasizes that it is a column vector of ones with dimension $L$.

$\square$

**For large argument**

As $\beta \to \infty$, the softmax function simply selects the maximum over its inputs (as long as the the maximum is unique):

$$\text{softmax}(\beta v) \approx \begin{cases} 1 & \text{if } i = \arg\max_j v_j, \\ 0 & \text{otherwise.} \end{cases}$$

In this case, all attention weight is given to a single element, and the others are effectively ignored.

## G. MSE Loss landscape for scaled identity weights

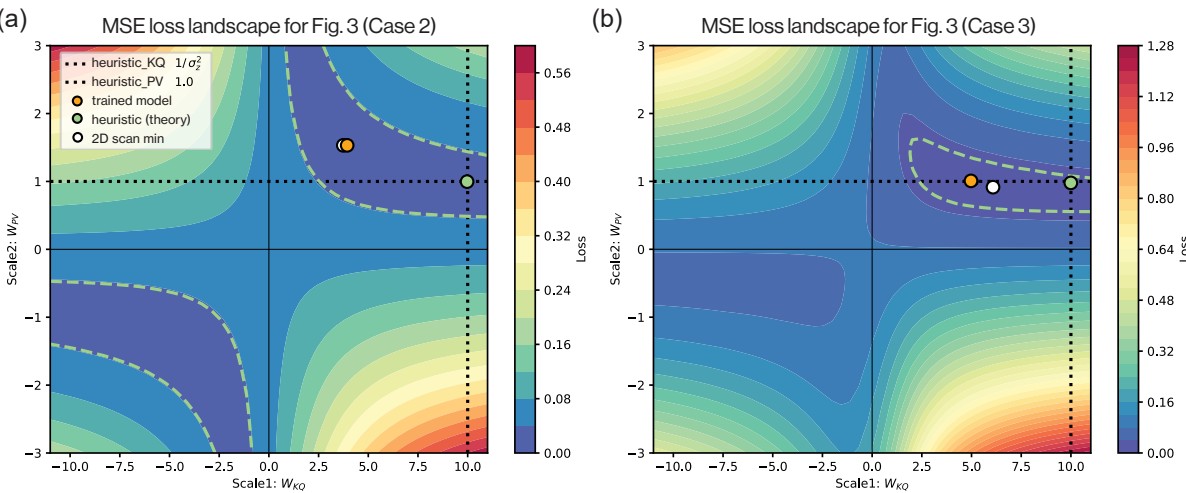

*Figure 6.* Loss landscape corresponding to Case 2 and Case 3 of Fig. 3. The MSE is numerically evaluated by assuming scaled identity weights $W_{KQ} = \beta I_n$ (x-axis) and $W_{PV} = \alpha I_n$ (y-axis) and scanning over a $50 \times 50$ grid. The green point corresponds to the heuristic minimizer identified from the posterior mean. In Case 2 it is exact, while in case 3 it is an approximation that neglects the residual term (see Proposition 4). The orange point corresponds to the learned weights displayed in Fig. 3(b), while the white point corresponds to the numerically identified minimum from this 2D scan. These can fluctuate due to the finite context ($L = 500$) and sampling ($N = 800$ here). In both panels, it is apparent that the trained weights and the heuristic estimator co-occur in a broad valley (contour) of the loss landscape.

The loss landscapes in Fig. 6 exhibit large, low-cost valleys with a roughly hyperbolic structure that is especially apparent in Case 2. This indicates a multiplicative tradeoff in the scales of $W_{KQ}$ and $W_{PV}$, which suggests that linear attention might be applicable here as well. For completeness, Figure 7 shows linear attention performance for both cases, demonstrating that it performs quite similarly to softmax for sub-sphere denoising, but less well in the Gaussian mixtures case.

## H. Structured optimal weights under prompt transformation

We find that one-layer transformers can learn to undo arbitrary invertible coordinate transformations that warp the denoising tasks. Focusing on the subspace denoising case, suppose each prompt is transformed by a fixed invertible square matrix

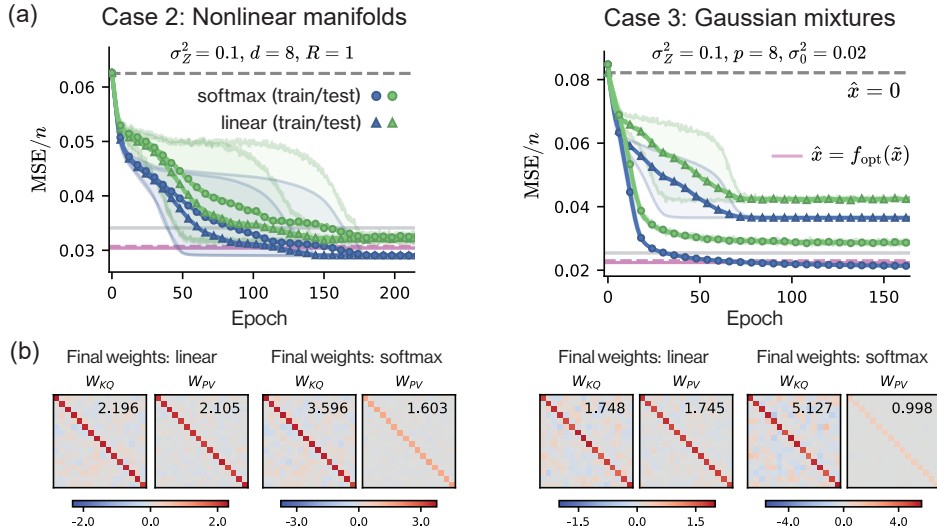

*Figure 7.* Linear attention performance for Cases 2 and 3. Additional empirical results for the nonlinear manifolds case (left) and the Gaussian mixtures case (right). (a) Loss dynamics for randomly initialized softmax and linear attention layers. Solid lines represent the average loss over six seeds, with shaded area indicating the range. Training details and parameters follow Fig. 3(a). (b) Representative final attention weights for each layer.

$A$, i.e. $E = (X_{1:L}, \tilde{x}) \rightarrow E' = (AX_{1:L}, A\tilde{x})$. If the target remains $x_{L+1}$ in the untransformed space, then the optimal attention weights are no longer diagonal, but instead take a structured form determined by the transformation matrix:

$$W_{PV} = \alpha A^{-1}, \quad W_{KQ} = \beta(AA^T)^{-1}, \tag{A.12}$$

where $\alpha\beta = \frac{1}{\sigma_0^2 + \sigma_Z^2}$ as before.

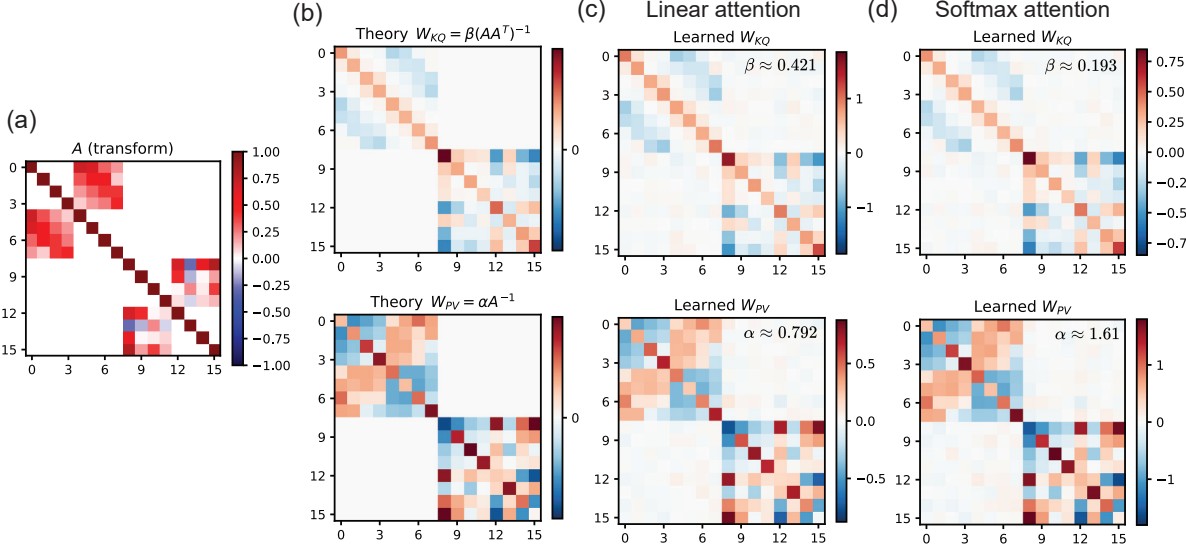

*Figure 8.* (a) Example transformation $A$ used to globally alter the in-context denoising prompts. (b) Structure of the optimal attention weights for this transformed subspace-denoising task. (c,d) Empirically, we find that both linear attention and softmax attention layers are able to learn these structured targets, but with distinct scalings $\alpha, \beta$. Final weights after 500 epochs using Adam, random initializations, and context length $L = 500$; other parameters follow Fig. 3(a).

Notably, we find that both the linear and softmax attention layers are able to learn these structures; see Fig. 8 for an example.

We use the same basic training procedure as the limiting case of $A = I$ (no additional coordinate transformation) assumed throughout the main text.

Suppose we still work with transformed coordinates $Y_t = AX_t$ and $\tilde{Y} = A\tilde{X}$, but now intend to retrieve $Y_{L+1} = AX_{L+1}$ in the new coordinate space (rather than $X_{L+1}$ as above). In this case, we would be dealing with variables with covariance matrices $\Sigma \propto AA^T$. We would need weight matrices that are not simply proportional to identity to deal with the covariance structure. This is also the case for in-context learning of linear functions when the input has an anisotropic covariance matrix (Zhang et al., 2024; Ahn et al., 2023). Recall in the original setting, we had the sample covariance $\mathbb{E}[XX^T] \equiv \Sigma_X = \sigma_0^2 P$ and noise $\Sigma_Z \equiv \sigma_Z^2 I$, leading to the estimator, Eq. (10): $\hat{X} = \frac{1}{(\sigma_0^2 + \sigma_Z^2)L} \sum_{t=1}^{L} X_t \langle X_t, \tilde{X} \rangle$. Here, the sample covariance is $\Sigma_Y \equiv \sigma_0^2 APA^T$, and the noise $V \equiv AZ$ has covariance $\Sigma_V \equiv \sigma_Z^2 AA^T$. One can show the generalized solution is $\hat{Y} = \Sigma_Y(\Sigma_Y + \Sigma_V)^{-1}\tilde{Y}$. Thus, in the transformed coordinates, the denoising estimate is

$$\hat{Y} = \frac{1}{(\sigma_0^2 + \sigma_Z^2)L} \sum_{t=1}^{L} Y_t \langle A^{-1}Y_t, A^{-1}\tilde{Y} \rangle. \tag{A.13}$$

For the relationship of this denoising result in $Y$ to energy models, as discussed in Section 4 and Subsection I.1, we need a modified energy $\mathcal{E}(Y_{1:L}, s) = \frac{1}{2\gamma}\|s\|^2 - \frac{1}{2L} \sum_{t=1}^{L} \langle A^{-1}Y_t, A^{-1}s \rangle^2$ and a preconditioner proportional to $AA^T$.

# I. Additional comments on the mapping from attention to associative memory models

## I.1. Linear attention and traditional Hopfield model

We have considered a trained network with linear attention, relating the query $\tilde{X}$ and the estimate of the target $\hat{X}$, of the form

$$\hat{X} = f(\tilde{X}) := \frac{\gamma}{L} \sum_{t=1}^{L} X_t \langle X_t, \tilde{X} \rangle \tag{A.14}$$

with $\gamma = \frac{1}{\sigma_0^2 + \sigma_Z^2}$.

With

$$\mathcal{E}(X_{1:L}, s) := \frac{1}{2\gamma}\|s\|^2 - \frac{1}{2L} s^T (\sum_{t=1}^{L} X_t X_t^T) s \tag{A.15}$$

gradient descent iteration $s(t+1) = s(t) - \gamma \nabla_s \mathcal{E}(X_{1:L}, s(t))$ gives us

$$s(t+1) = \frac{\gamma}{L} \sum_t X_t \langle X_t, s(t) \rangle$$

making the one-step iteration our denoising operation.

We will call this energy function the Naive Spherical Hopfield model for the following reason. For random memory patterns $X_{1:L}$, and the query denoting Ising spins $s \in \{-1, 1\}^n$, the so-called Hopfield energy is

$$\mathcal{E}_{\text{Hopfield}}(X_{1:L}, s) := -\frac{1}{2L} s^T (\sum_{t=1}^{L} X_t X_t^T) s. \tag{A.16}$$

We could relax the Ising nature of the spins by letting $s \in \mathbb{R}^n$, with a constraint $\|s\|^2 = n$. This is the spherical model (Fischer & Hertz, 1993) since the spin vector $s$ lives on a sphere. If we minimize this energy the optimal $s$ would be aligned with the dominant eigenvector of the matrix $\frac{1}{L}(\sum_{t=1}^{L} X_t X_t^T)$ (Fischer & Hertz, 1993), and the model will not have a retrieval phase (see Bollé et al. (2003) for a similar model that does). A soft-constrained variant can also be found in Section 3.3, Model C of Krotov & Hopfield (2021).

We could reformulate the optimization problem of minimizing the Hopfield energy, subject to $\|s\|^2 = R^2$, as

$$\underset{s \in \mathbb{R}^n}{\arg\min} \left[ \max_{\lambda} \left\{ -\frac{1}{2L} s^T (\sum_{t=1}^{L} X_t X_t^T) s + \lambda(s^T s - R^2) \right\} \right].$$

The $s$-dependent part of the Lagrangian, with $\lambda$ replaced by $\frac{1}{2\gamma}$ gives us the energy function in Eq. A.15 which we have called the Naive Spherical Hopfield model.

$$\mathcal{E}(X_{1:L}, s) := \frac{1}{2\gamma}\|s\|^2 - \frac{1}{2L}s^T(\sum_{t=1}^{L}X_tX_t^T)s = \frac{1}{2}s^T\Big[(\sigma_0^2 + \sigma_Z^2)I_n - \frac{1}{L}(\sum_{t=1}^{L}X_tX_t^T)\Big]s. \qquad (A.17)$$

For $L$ much larger than $n$, $\frac{1}{L}\sum_{t=1}^{L}X_tX_t^T \approx \sigma_0^2 P$, so its eigenvalues are either 0 or are very close to $\sigma_0^2$. Hence, for large $L$ and $\sigma_Z > 0$, this quadratic function is very likely to be positive definite. One-step gradient descent brings $s$ down to the $d$-dimensional linear subspace $S$ spanned by the patterns, but repeated gradient descent steps would take $s$ towards zero.

### I.2. Remarks on the softmax attention case (mapping to dense associative memory networks)

Regarding the mapping discussed in the main text, we note that there is a symmetry condition on the weights $W_{KQ}, W_{PV}$ that is necessary for the softmax update to be interpreted as a gradient descent (i.e. a conservative flow). In general, a flow $ds/dt = f(s)$ is conservative if it can be written as the gradient of a potential, i.e. $f(s) = -\nabla_s V(s)$ for some $V$. For this to hold, the Jacobian of the dynamics $J_f(s) = \nabla_s f$ must be symmetric.

The softmax layer studied in the main text is $f(s) = W_{PV}X\,\mathrm{softmax}(X^T W_{KQ}s)$. We will denote $z(s) = X^T W_{KQ}s$ and $g(s) = \mathrm{softmax}(z(s))$, both in $\mathbb{R}^L$. Then the Jacobian is

$$J(s) = W_{PV}X\frac{\partial g}{\partial s} = W_{PV}X\left(\mathrm{diag}(g) - gg^T\right)X^T W_{KQ}. \qquad (A.18)$$

Observe that $Y = X\left(\mathrm{diag}(g) - gg^T\right)X^T$ is symmetric (keeping in mind that $g(s)$ depends on $W_{KQ}$). The Jacobian symmetry requirement $J = J^T$ therefore places the following constraint on feasible $W_{KQ}, W_{PV}$:

$$W_{PV}\,Y\,W_{KQ}^T = W_{KQ}\,Y\,W_{PV}^T. \qquad (A.19)$$

It is clear that this condition holds for the scaled identity attention weights discussed in the main text. Potentially, it could allow for more general weights that might arise from non-isotropic denoising tasks to be cast as gradient descent updates.

The mapping discussed in the main text involves discrete gradient descent steps, Eq. (17). In general, this update rule retains a "residual" term in $s(t)$ if we choose a different descent step size $\gamma \neq \alpha$. Thus, taking $K$ recurrent updates could be viewed as the depthwise propagation of query updates through a $K$-layer architecture if one were to use tied weights. Analogous residual streams are commonly utilized in more elaborate transformer architectures to help propagate information to downstream attention heads.

