# OpenReview forum: "In-Context Denoising with One-Layer Transformers: Connections between Attention and Associative Memory Retrieval"
_ICML.cc/2025/Conference — ICML 2025 oral_

### Official Review · Reviewer_BQoS · 2025-03-07

**Overall Recommendation:** 4

**Summary:**

## Update After Rebuttal
## I maintain my score. Please see below for my reasonings in my response. Overall, I think it is a great paper despite some of my comments.
-----
This work introduces a concept, in-context denoising, which is a task that refines the connection between attention-based architectures and Dense Associative Memory networks (DAMs). Using a Bayesian framework, based on the minimization problem of $\mathbb{E}_{X, \tilde{X}} [X - f_\theta(\tilde{X})]$, where $\tilde{X}$ is the perturbed version of $X$, this paper illustrates that certain context-dependent denoising problems can be solved with a single-layer transformer model. Furthermore, it also demonstrates that a context-aware DAM one-step update yields better performance than a non-context-aware DAM update. Altogether, the work further solidifies the connection between attention and associative memory (AM), while illustrating the relevancy of AM for in-context learning (ICL).

**Claims And Evidence:**

### Claim

The paper has two fundamental claims:

(1) They claim that there are certain denoising tasks that a single-layer transformer can effectively solve.

(2) Once trained, the attention layer effectively performs an operation that is a single gradient descent step on a context-aware DAM's energy landscape. This update is better than an extract retrieval of a token or spurious sample.

### Evidence

(1) To support the first claim, the presents theoretical results on three elementary denoising tasks where the data comes from (a) linear manifolds, (b) non-linear manifolds and (c) Gaussian mixtures or clusters, see Figure 1. Specifically, it presents the Bayesian optimal function $f_\text{opt}$ for each of these cases. For each case of data, the work makes an argument that the $f_\text{opt}$ of each case is equivalent to the attention mechanism of different activation (e.g., linear or softmax attention).

To further support their statement on $f_\text{opt}$ of each case being equivalent to the attention mechanism. They perform experiments illustrated in Figure (3), which highlights the MSE loss of the estimated $\hat{x}$ and the expected loss obtained from $f_\text{opt}$. This overall demonstrates that the 1-layer transformer model will converge to the expected loss obtained from the Bayesian optimal predictor. Note, this particular experiment is pretty convincing.

(2) The connection between DAM and attention is rather straightforward since [Ramsaeur et al. (2020)](https://arxiv.org/abs/2008.02217) has already established this initially. To further establish this connection, this work connects the DAM energy to ICL where the context sequence $X_{1:L}$ serves as the memory in which the noised prompt $\hat{x}$ uses for its alignment with the system. However, this connection is quite brief and obvious, since the formulation is already done [Ramsaeur et al. (2020)](https://arxiv.org/abs/2008.02217).

To support their second claim, they perform a simple experiment detailed in Figure 5. In this experiment, using the DAM energy equation, which involves the term $\frac{1}{2 \alpha} \lVert s \rVert^2$, they contrast the update rule
$s(t + 1) = (1 - \frac{\gamma}{\alpha}) s(t) + \gamma \, X_{1:L} \, \text{softmax}(\beta X_{1:L}^\top s(t))$. Specifically, they contrast when the update rule is equivalent to the attention mechanism, i.e., $\gamma = \alpha$, and when $\gamma \neq \alpha$. As already shown in [Ramsaeur et al. (2020)](https://arxiv.org/abs/2008.02217), when $\gamma = \alpha$, the retrieval is fast and accurate and shown here by this work, when $\gamma \neq \alpha$, the retrieval diverges away from the target point because the dynamics become query-independent.

## Strength

(1) The theoretical results regarding $f_\text{opt}$ for the three data cases presented in this paper are concrete and easy to understand. The paper also presents the expected optimal weight for the single-layer transformer once it is trained. This weight is simply a scaled identity matrix.

(2) Another interesting result presented in the paper is the effect of increasing context length on the convergence to the expected bound based on the Bayesian optimal predictor. With sufficient (or increasing) context length, it seems that the convergence rate is increased.

(3) This paper provides an interesting in the dynamics of transformer regarding denoising problem and further connects such dynamics to DAM.

## Weaknesses

(1) The analyses are done on a single-layer transformer system. It is unclear on how normal models, i.e., those with a large depth, behaves. Specifically, how does their weights vary and what effects do they play in performing multiple iterative steps?

(2) The connection to DAM feels a bit minimal, which is somewhat understandable since much of the results are focused on the theoretical section involving the three general denoising cases that a simple transformer can solve.

(3) Although the one-step update in DAM (i.e., when $\gamma = \alpha$) is faster to converge or accurate in terms for the retrieval setting, it's not exactly better depending on the task. For example, in terms of denoising to generate new data, the update with $\gamma \neq \alpha$
could be better, since you don't need exact retrieval in this case.

**Essential References Not Discussed:**

A nice reference that could be mentioned is the Energy Transformer (or [ET](https://arxiv.org/abs/2302.07253)). It is an extension of [Krotov and Hopfield (2021)](https://arxiv.org/abs/2008.06996) and [Ramsaeur et al. (2020)](https://arxiv.org/abs/2008.02217) which converts the non-dynamical Transformer block into a dynamical (Energy) Transformer block.

**Experimental Designs Or Analyses:**

As mentioned prior, I see no flaws in the experimental designs and their analyses. My general issues are laid out in the weaknesses above.

**Methods And Evaluation Criteria:**

(1) The methods or experimentation which evaluated claim (1) make a lot of sense. The authors trained a single-layer transformer and analyze whether its convergence is bounded by their theoretical value. I don't see faults on their experimentation.

(2) The experimentation for claim (2) is straightforward and quite simple. Once again, there are no faults in their approach here.

**Other Comments Or Suggestions:**

(1) I think there are a lot of results presented in the paper, which is a strength. However, these results can be quite complex for some readers. I think leading the presentation of the paper with the narrative of DAM can help keep the interests of the readers.

(2) In section (C.2) of Appendix C, there is an incomplete sentence at the end of the sub-section:

"The clustering case"

**Other Strengths And Weaknesses:**

N/A

**Questions For Authors:**

See my weaknesses above.

**Relation To Broader Scientific Literature:**

In general, we now know that the attention mechanism from [Vaswani et al. (2017)](https://arxiv.org/abs/1706.03762) is connected to the Modern Hopfield Model, developed by [Krotov and Hopfield (2016)](https://arxiv.org/abs/1606.01164) and [Demircigil et al. (2017)](https://arxiv.org/abs/1702.01929), shown by [Ramsaeur et al. (2020)](https://arxiv.org/abs/2008.02217). Moreover, we know how powerful the applications developed with the Transformer architecture can be, e.g., ViTs [Dosovitskiy et al.](https://arxiv.org/abs/2010.11929). ***But we still don't know why the transformer block works so well and what exactly does the mechanism of the block means***.

With the perspective of DAM, we know the attention mechanism is simply a gradient step on the DAM's energy landscape as shown by [Ramsaeur et al. (2020)](https://arxiv.org/abs/2008.02217) and further reinforced and explained in this work. With simple data cases, this work demonstrates that a trained single-layer attention model will converge to become a gradient step on a Hopfield model's energy landscape. This is quite significant.

**Theoretical Claims:**

I (the reviewer) primarily paid much of my attention in sections (2 and 3) since they contained the theoretical proof and results. I don't see any problems with the proofs nor the claims. My general issues are laid out in the weaknesses above.

---

> ### Author Rebuttal · Authors · 2025-03-31
>
> We thank the reviewer for their positive assessment and the high score (4/5) provided. We concur with their observation that, collectively, we still don't know why the transformer block works so well. This shared curiosity drives our work to build theoretical foundations for understanding the mechanisms underlying the success and flexibility of attention mechanisms. We are thus deeply grateful for their positive feedback in this direction.
>
> **Re: Claims and Evidence**
> We respectfully disagree with point (2) in the evidence section. While our paper builds on the connection established by Ramsauer et al., our contribution goes significantly beyond this foundation. Reading Ramsauer et al., one gets the impression that the question is memory retrieval. In our formulation, the task could be quite distinct from memory retrieval. We set up an in-context learning task, whose one-layer transformer-based solution is not guaranteed to look like a one-step lowering of some simple energy function. Quite remarkably, it does so in the special cases we study.
>
> **Re: Essential References**
> We thank the reviewer for highlighting the Energy Transformer (ET) paper and regret missing the citation. It is distinct but complementary effort that will be discussed in our revised manuscript to better situate our contribution within the broader literature.
>
> **Re: Strengths and Weaknesses**
> Regarding the weaknesses discussed:
> - (1) We agree that analyzing deeper transformer systems is an important direction. Our focus on single-layer transformers was deliberate to establish clear connections in this foundational case. The question of how weights in deeper networks might implement multiple iterative steps on energy landscapes represents an exciting avenue for future research.
> - (2) This paper was written as an invitation to those working in mainstream transformer theory towards the very interesting activity in the DAM community. The in-context denoising task was meant to be a bridge between these two worlds. We will include further discussion on connections to DAM in the final version, if accepted, such as a detailed comparison with the Energy Transformer and a discussion of how our approach relates to tasks of interest in the DAM community, which we initially omitted due to space limitations.
> - (3) Indeed, in specific retrieval tasks, one might very well benefit from multiple steps or varied step sizes. In our specialized denoising cases, the 'step size' gets fixed during training, where the transformer is trying to approximate the Bayes optimal answer. This highlights an interesting distinction between retrieval tasks and denoising tasks that could inform broader transformer design.
>
> **Re: Comments and Suggestions**
> - (1) Thanks for the kind words! We are certainly open to refocusing the introduction to keep the interest of readers, but would prefer to lead the results section with in-context learning and the Bayesian setting + derivations, as we feel it sets the stage well for the emergence of the associative memory behavior in the optimal/trained networks.
>
> - (2) Thanks much for catching the incomplete sentence at the end of subsection C.2 in the Appendix. It was meant to be part of the header for Section C.3. We will correct this in the revised version.
>
> --
>
> We thank the reviewer again for their positive feedback and valuable suggestions. We believe that addressing these points will further strengthen the paper while maintaining its core contributions to solidifying the connection between attention mechanisms and associative memory in a minimal in-context learning setting.

---

> > ### Comment · Reviewer_BQoS · 2025-04-01
> >
> > To the authors,
> >
> > Thank you for your response. Overall, I enjoy reading your paper. Although, my opinion could be wrong, I still feel the connection to DAM is brief. However, I believe the results presented in the paper are adequate and tell a nice story about ICL and DAM. Another thing that could be mentioned regarding Eqs. (15 and 16) is that --- if the update rule only involves the softmax term (i.e., $\gamma = \alpha$), we get a very nice update rule for memory retrieval but when $\gamma \neq \alpha$, this could actually be perfect for generation or synthesizing new minima from the stored memories (if that makes sense).
> >
> > A few more suggestions from me are:
> >
> > (1) fix line (642-643)
> >
> > (2) fix line (674-675)
> >
> > (3) perhaps move Fig. (6) to the main text given the additional page for camera-version
> >
> > (4) increase the size of Fig. 5 --- the font size of the legend/labels seems too small to me
> >
> > (5) Fig. 4 could be stretch out a bit more, I believe if you crop out the white space of that figure, Latex should be able to make it larger.
> >
> > Overall, I would like to maintain my score.
> >
> > Best of luck to the authors.

---

> > > ### Author Response · Authors · 2025-04-08
> > >
> > > We thank the reviewer for their positive feedback and helpful suggestions towards improving the work. We appreciate the insightful idea about the update rule when $\gamma \neq \alpha$ potentially being useful for generation, as well as the formatting recommendations. We will incorporate these points in our camera-ready version if accepted.

---

### Official Review · Reviewer_kZ23 · 2025-03-11

**Overall Recommendation:** 4

**Summary:**

This paper studies the in-context unsupervised denoising of data points in transformers. They show that single layer transformers with a single attention head are sufficient to learn this task and that standard training procedures from random initialization can converge to Bayes optimal solutions. Lastly, they provide connections between the learned transformer models on this task and Dense Associative Memory models, a modern variant of Hopfield networks which have high storage capacity.

**Claims And Evidence:**

The authors provide ample theoretical and empirical support of their claims. They provide derivations of all of the Bayes optimal predictions $\mathbb{E}[X|\tilde{X}]$ for the various settings they consider, they provide constructions which show that transformers can implement these solutions, and they also provide many experiments where they train transformers on these tasks and show the ability of the model to achieve Bayes optimal error.

**Essential References Not Discussed:**

Possibly of interest is this work https://arxiv.org/abs/2310.09753 which finds that adding additional identity matrix to the product $W_k^\top W_Q$ can lead to improved performance on template "reasoning" tasks. This reminds me somewhat of the fact that you argue that $W_k^\top W_Q$ should be close to a scaled identity in this work.

**Experimental Designs Or Analyses:**

I did not check the experimental design in great detail, but I trust the simulations.

**Methods And Evaluation Criteria:**

Yes, the paper primarily focuses on synthetic data but this is the setting of interest for their theory.

**Other Comments Or Suggestions:**

N/A

**Other Strengths And Weaknesses:**

This paper studies an interesting problem, gives precise and testable predictions and verifies them. One weakness at the present moment is the limited scope of data distributions that are considered, but I think that the present contribution is already good.

**Questions For Authors:**

1. The authors provide mainly asymptotic results as context length diverges $L \to \infty$. Do they have any sense of the convergence rate in $L$ as a function of the dimension of the space? Specifically how large does $L$ need to be for the finite sample sums in equation 11 or 13 need to be to converge to the correct estimate?
2. The construction concluded that $W_k^\top W_q$ should be a scaled version of the identity, but in experiments it doesn't have to be exactly this. The authors then provide a short discussion on why this could be, specifically that there are different values of $W_k^\top W_q$ and $W_v$ that implement equivalent attention operations. Have the authors checked if in fact the learned implementation matches their theoretical expectations?

**Relation To Broader Scientific Literature:**

This paper studies an important problem of in-context capabilities of transformers, specifically in the context of denoising. They show that transformers can theoretically and empirically learn to denoise several distributions. In addition, they also provide interesting connections to the associative memory literature.

**Theoretical Claims:**

Yes, I checked the theoretical claims, most of which are easy to verify by following the derivations in the Appendix.

---

> ### Author Rebuttal · Authors · 2025-03-31
>
> We appreciate the reviewer's thoughtful review and their high evaluation of our paper (4/5). We agree that this is an important problem and are thus grateful for their positive assessment of our theoretical findings and empirical validations.
>
> **Re: Weaknesses**
> The reviewer notes that our paper primarily focuses on synthetic data. This was a deliberate design choice to establish clear theoretical connections between transformer attention and Dense Associative Memory (DAM) networks. By working with well-defined synthetic distributions, we could derive Bayes-optimal predictors as theoretical baselines against which to evaluate our transformer models.
>
> We agree that extending these insights to more complex, real-world data distributions represents an important direction for future work. The current paper focuses on establishing foundational theoretical results and validating them empirically in controlled settings where optimality can be rigorously established. This approach allows us to identify the key mechanisms by which transformer attention implements optimal denoising, providing insights that can inform future work on more complex data distributions.
>
> **Re: References**
> We thank the reviewer for pointing us to the work of Boix-Adsera et al. This paper's finding that adding an identity matrix to the product $W_K^T W_Q$ can improve performance on template reasoning tasks (Observation 1.2) resonates with our work. We will include this reference in our revision (perhaps alongside Trockman & Kolter (2023) in the discussion) to highlight this potential connection and suggest avenues for future investigation.
>
> **Re: Specific Questions**
> - **(Q1)** Regarding the convergence rate in $L$ as a function of dimension: we appreciate this question and will consider including a more detailed analysis of finite-sample effects in the Appendix. Briefly, our analysis primarily focused on the asymptotic behavior as $L \rightarrow \infty$ (using the strong law of large numbers, which just requires the mean to exist). However, in the linear example, our tokens are Gaussian and in the two non-linear cases they are bounded. Intuitively, we expect error $O(\sqrt\frac{1}{L})$. In fact, we can give precise probabilistic bounds of the form that the difference between the empirical sum for the ideal weights depart from the expectation by less than $C(\tilde x)\sqrt{\frac{f(d,\ln\frac{1}{\delta})}{L}}$ is greater than $1-\delta$. The function $C$ of the query vector and the function $f$ depends on the problem. Interestingly, this indicates $d$ depends on the dimension spanned by the tokens (not the ambient dimension $n$). Figure 4(a) provides some empirical evidence for this relationship, showing how performance improves with increasing context length.
> - **(Q2)** Regarding the learned implementation matching theoretical expectations: Yes, we have verified this carefully. Figure 3(b) shows the final learned attention weights, which closely resemble scaled identity matrices as predicted by our theory. We also analyzed the loss landscape with respect to scaling factors $\alpha$ and $\beta$, confirming that the learned weights indeed lie in the predicted valley of the loss landscape. The minor deviations from the exact theoretical values can be attributed to finite context effects and the stochastic nature of the training process, but do not contradict our theoretical findings.
>
> --
>
> We thank the reviewer again for their thoughtful review and insightful questions. Their suggestions have helped us identify important points to clarify and highlight in the final version of our paper.

---

> > ### Comment · Reviewer_kZ23 · 2025-04-03
> >
> > I thank the authors for their detailed comments and maintain my positive evaluation.

---

> > > ### Author Response · Authors · 2025-04-08
> > >
> > > We thank the reviewer again for their positive feedback. We will try our best to incorporate their suggestions in the camera-ready version if accepted.

---

### Official Review · Reviewer_RPZT · 2025-03-13

**Overall Recommendation:** 3

**Summary:**

The paper explores a link between one-layer transformer attention and associative memory retrieval.
It frames an in-context denoising setting with three synthetic data scenarios: linear subspaces, points on a sphere, and Gaussian mixture clusters.
It derives Bayes-optimal predictors in each setting and shows that a single attention update approximates these optimal solutions under small-scale assumptions.
It interprets the one-layer transformer as one-step gradient descent on a Hopfield-like energy. Iterative updates converge to stored attractors and degrade denoising performance.
The paper supports these claims with synthetic experiments, where trained transformers match the predicted optimal performance.
 It concludes that a one-step attention update implements Bayes-optimal denoising in specific synthetic contexts, connecting transformer attention with associative memory beyond exact pattern retrieval.

## update after rebuttal

**Claims And Evidence:**

* **Claim: Single-step attention approximates Bayes-optimal denoisers in specific synthetic settings**
    * Evidence: The authors derive exact optimal predictors under Gaussian mixtures, linear subspaces, and spherical assumptions. They show near-optimal mean squared error in experiments.
* **Claim: One-layer transformers match a one-step gradient update on a Hopfield-like energy**
    * Evidence: The paper presents an energy formulation and demonstrates that learned attention weights align with the gradient of that energy. Experimental training runs confirm that a single attention update outperforms multiple updates in denoising.

**Essential References Not Discussed:**

-

**Experimental Designs Or Analyses:**

**Conclusion:** The experimental design is well aligned with the theory. The analyses are correct for the stated goals. The limited scope leaves open questions about broader generalization.


---

**Synthetic Task Design:** The authors focus on three artificial data setups: linear subspaces, points on a sphere, and Gaussian mixture clusters. They construct these tasks to mirror the paper’s theoretical assumptions (e.g., isotropy, symmetry). This design is consistent with the paper’s goal of checking Bayes-optimality in simplified contexts.

**Model Configuration:** They use one-layer transformers with either linear or softmax attention. They match each architecture to a theoretical predictor. The design supports the core claim that a single-step attention update aligns with the derived solutions.

**Training Procedure:** They train these models on the synthetic tasks and measure mean squared error, checking how close each model’s performance is to the analytical Bayes-optimal denoiser. This procedure is valid. They compare different initializations and sometimes vary noise scales to confirm robustness.

**Comparisons:** They contrast linear attention versus softmax attention. They show that both approaches achieve near-Bayes-optimal performance. They analyze how the learned attention weights resemble the predicted identity-like matrix in each scenario.

**Limitations:** They do not test real-world data or more complex tasks. The synthetic setups are narrow in scope. I guess this is fine for a theoretical paper.

**Methods And Evaluation Criteria:**

-

**Other Comments Or Suggestions:**

see above

**Other Strengths And Weaknesses:**

## Strengths

- **Originality:**
  Introduces in-context denoising as a new setting, explicitly linking transformer attention to Hopfield memory. Extends previous views that only consider retrieval tasks.

- **Significance:**
  Clearly demonstrates the significance of transformers performing optimal denoising updates beyond single-pattern retrieval, providing a formal connection between in-context attention and optimal memory retrieval.

- **Clarity:**
  Clearly explains the proposed approach and its advantages, with logical connections between theoretical analysis and experiments.

## Weaknesses

- **Positioning and Contribution:**
  The main concern is the unclear positioning of the paper’s contribution. Transformers implicitly simulate gradient-based updates in-context, as established by existing literature. Prior studies already demonstrate this perspective, though typically limited to single-token retrieval.

    After some reflections, I suggest positioning this paper as extending the known "transformers-as-implicit-gradients" concept. Through 3 very specialized examples, this paper analyzes how attention interpolates among multiple context tokens to achieve a Bayes-optimal denoiser, rather than merely retrieving or copying one pattern. Specifically, this paper goes beyond prior studies by:

  - Performs optimization blending multiple context tokens, achieving performance superior to single-token retrieval.
  - Demonstrates Bayes-optimality explicitly in simplified synthetic tasks.
  - Clarifies how mixed retrieval outperforms exact retrieval, providing rigor to the implicit-gradient viewpoint in multi-token scenarios.

    This adds a conceptual and theoretical layer to the implicit GD story: it clarifies why partial or “mixed” retrieval can be optimal, whereas previous examples focused on simpler single-token or exact retrieval scenarios.

- **Literature Completeness:**
  Several relevant references about the association between attention mechanisms and one-step associative memory retrieval are missing. Including these references improves the contextual completeness of the paper.

- **Equation referencing:**
  Some equations have numbers without explicit references in the main text. Equations that are not explicitly cited should not have numbers.

## Why not higher score?

- The paper primarily refines known connections between transformers and Hopfield networks. The core novelty is modest, building on established frameworks rather than providing fundamental insights.

- The theoretical analysis and experiments focus exclusively on simplified synthetic scenarios, limiting generalization to more realistic or practical applications.

- The limited experimental and theoretical scope reduces the broader impact. Expanding experiments or theoretical analysis beyond synthetic settings significantly strengthens this work.

**Questions For Authors:**

1. whether the one-step Bayes-optimal property holds when data are more complex, structured, or high-dimensional (e.g., text or images)? for example, how read the corresponding attention be like? is it possible to be sth like really attention heads in practice?

2. What happens if we are allowed to perform multiple iterative updates? Do additional steps help or hinder performance?

3. Does the Bayes-optimality of one-step updates hold in 1-layer multi-head attention?

**Relation To Broader Scientific Literature:**

### **Connection to Modern Hopfield Networks:**

The paper builds on the established link between transformer attention and modern Hopfield retrieval (Ramsauer et al., 2021). It interprets a single-layer transformer as a one-step gradient update on a Hopfield-like energy. Similar efforts in recent literature but not mentioned in this paper include
-  [Santos et al. ICML 2024] Sparse and Structured Hopfield Networks https://arxiv.org/abs/2402.13725
-  [Wu et al., ICML 2024] Uniform Memory Retrieval with Larger Capacity for Modern Hopfield Models https://arxiv.org/abs/2404.03827
-  [Wu et al., ICLR 2024] STanHop: Sparse Tandem Hopfield Model for Memory-Enhanced Time Series Prediction https://arxiv.org/abs/2312.17346
-  [Hu et al., NeurIPS 2023] On Sparse Modern Hopfield Model https://arxiv.org/abs/2309.12673

The paper’s one-step approach complements these ideas but does not discuss them.

---

### **Extension from Exact Retrieval to Denoising:**

Prior Hopfield models often retrieve exact stored patterns, e.g., [Santos et al. ICML 2024]. The paper shifts toward Bayes-optimal denoising, which involves partial or mixed retrieval rather than convergence to a single pattern.

---

### **Broader Transformers and In-Context Learning:**

The paper adds to literature suggesting transformers perform implicit gradient updates in-context (for examples, see below). Prior works demonstrate in-context meta-learning [Dai et al., ACL 2023] or approximate Bayesian inference [Xie et al., ICLR 2022] , and this paper contributes a denoising viewpoint.

- [Oswald et al., ICML 2023] Transformers learn in-context by gradient descent
 https://arxiv.org/abs/2212.07677
- [Dai et al., ACL 2023] Why Can GPT Learn In-Context? Language Models Implicitly Perform Gradient Descent as Meta-Optimizers https://arxiv.org/abs/2212.10559
- [Bai et al., NeurIPS 2023] Transformers as Statisticians: Provable In-Context Learning with In-Context Algorithm Selection https://arxiv.org/abs/2306.04637
- [Ahn et al., NeurIPS 2023] Transformers learn to implement preconditioned gradient descent for in-context learning https://arxiv.org/abs/2306.00297
- [Xie et al., ICLR 2022] An Explanation of In-context Learning as Implicit Bayesian Inference
 https://arxiv.org/abs/2111.02080

**Theoretical Claims:**

All key claims are backed by formal derivations (with detailed proofs in the appendices). The math seems sound. For example, Proposition 2–4 give closed-form optimal denoisers for each case, and Proposition 3.2 analyzes the softmax attention in a small-argument regime. These ensure the theoretical claims hold under certain limits.

Throughout, assumptions (e.g. isotropic distributions, large ambient dimension, small noise limits) are clearly stated, and the derivations use standard Bayesian inference and asymptotic analysis, which appear logically valid.

---

> ### Author Rebuttal · Authors · 2025-03-31
>
> We appreciate the reviewer's thoughtful engagement with our work, particularly their recognition of our work's **originality** in studying in-context denoising, its **significance** in demonstrating transformers performing optimal denoising updates beyond exact pattern retrieval, and its **clarity** in explaining the proposed approach. We believe these findings establish fundamental connections between attention architectures and Dense Associative Memory networks through Bayes-optimal denoising -- providing insights that extend beyond incremental advances while also establishing a bridge to the study of in-context learning.
>
> **Re: Positioning and Contribution**
> Our paper makes a specific contribution beyond prior work connecting transformers and Hopfield networks: establishing that for certain denoising tasks, a single attention step can be optimal from a Bayesian perspective. This provides theoretical justification for why the "one-step correspondence" noted by Ramsauer et al. (2020) works effectively. Our key insight is that in-context denoising connects associative memory retrieval, attention, and ICL in a common Bayesian framework. We demonstrate that:
> - For certain denoising tasks, one-layer transformers can represent Bayes-optimal predictors
> - Trained attention layers correspond to single gradient descent steps on a context-dependent energy landscape
> - This single step can outperform multiple iterations, challenging conventional associative memory paradigms. This extends beyond the common view that associative memories must converge (often through multiple iterations) to be effective.
>
> **Re: Relation To Broader Scientific Literature**
> We thank the reviewer for highlighting important related works. We will incorporate these references, particularly Santos et al. (ICML 2024), Wu et al. (ICML 2024), Wu et al. (ICLR 2024), and Hu et al. (NeurIPS 2023) on sparse modern Hopfield networks. We'll also discuss connections to implicit gradient descent literature mentioned by the reviewer. These works complement our findings while our contribution on optimal one-step denoising adds a new perspective.
>
> **Re: Equation referencing**
> We are open to doing so (especially if it is an ICML guideline), but regard it as a style choice, as readers may mention specific equations even if they aren't referenced within the text.
>
> **Re: Strengths and Weaknesses**
> Regarding our energy function differing from standard DAM: This modification was deliberate and essential for our denoising tasks. The Lagrange multiplier term handles continuous state spaces more naturally while maintaining core associative memory dynamics, similar to regularization approaches in other energy-based models.
>
> As for additional iterations degrading performance: This is a key finding, not a limitation. Traditional DAMs aim to retrieve patterns exactly, but in-context denoising requires blending information from context tokens with the corrupted query. Figure 5 demonstrates why a single step is optimal for certain denoising tasks, providing theoretical insight into when one-step updates outperform iterative approaches.
>
> **Re: Why not higher score?**
> We respectfully suggest reconsidering our contribution:
>
> - Our paper provides fundamental insights into why single-step attention can be optimal for denoising tasks, moving beyond merely refining known connections and establishing the ICL connection.
> - Our focus on simplified synthetic scenarios establishes rigorous theoretical foundations, a common approach in theoretical ML papers.
> - The impact extends beyond specific scenarios studied. By establishing when single-step updates can be optimal, we provide insights informing design and analysis of more complex transformer architectures.
>
> **Re: Questions for Authors**
> - **(1) & (3):** We've shown that for elementary denoising tasks, the optimal solution is a one-layer, single-head attention architecture viewable through the DAM lens. For more complex tasks, multi-layer/multi-head architectures likely become necessary for Bayes-optimality -- an exciting direction for future research.
> - **(2):** This important question motivated Section 4. Traditional views suggest convergence via iterative attention updates are beneficial. Fig. 5 shows how one-step outperforms multiple iterations in our denoising task. Multiple steps push the query toward fixed points that depend on random sampling, causing the system to 'forget' query information - sub-optimal for denoising. This insight reinforces the utility of the single attention step: it strikes an optimal balance between query and context information. We will discuss this in the revised text.
>
> --
>
> The reviewer's thoughtful feedback and insightful questions are much appreciated. We believe that addressing these points will strengthen the paper while maintaining its core contribution: solidifying the connection between attention mechanisms and associative memory through a novel in-context denoising setting.

---

### Official Review · Reviewer_PjG9 · 2025-03-13

**Overall Recommendation:** 4

**Summary:**

The paper considers in-context denoising as a fundamental task of attention (when applied in a prompt-conditioned, auto-regressive manner). When interpreted this way, there are clear connections to Dense Associative Memories (DAMs), and a model trained on a final-token denoising task can approach the optimal Bayesian prediction when knowledge of the true manifold is known in advance.

**Claims And Evidence:**

Claims are supported by simple experiments and theoretical analyses.

**Essential References Not Discussed:**

A very related but undiscussed work is the [Energy Transformer](https://arxiv.org/abs/2302.07253), which formalizes the entire Transformer block as a kind of DAM. Specifically, the paper contains several statements that are explicitly addressed by the Energy Transformer:

> [L068-070] why does the Ramsauer et al. (2021) correspondence involve only one iteration of Hopfield energy minimization and not many?

> [L397-399] We have also noted preliminary connections between our work and other architectural features of modern transformers, namely layer normalization and residual streams, which warrant further study.

> [L949-952] taking K recurrent updates could be viewed as the depthwise propagation of query updates through a K-layer architecture if one were to use tied weights. Analogous residual streams are commonly utilized in more elaborate transformer architectures to help propagate information to downstream attention heads.

Additionally, the work of [Ambrogioni](https://arxiv.org/abs/2309.17290) should be mentioned alongside [Hoover et al.](https://arxiv.org/abs/2309.16750) on [L412]

**Experimental Designs Or Analyses:**

The experimental designs are sufficient to test the claims in this work.

**Methods And Evaluation Criteria:**

The methods and evaluations are sound. It would be nice to see experiments on real data, not only toy data for which we know the underlying distribution $p_\mathcal{X}$, see weaknesses.

**Other Comments Or Suggestions:**

**Typos**

- Is there an error in Appendix E's taylor expansion of the softmax [L814] and [L822]? It seems like a math symbol did not render correctly.

**Other Strengths And Weaknesses:**

### Strengths
- **Well written**. Clear exposition and motivations makes this paper a pleasure to read. The figures are generally complete and self-descriptive (though a few details can be clarified, see questions).
- **Strongly defended connection of ICL to Dense AM**. The idea that ICL (and indeed, much of the operation of transformers) can be studied through the perspective of associative memory and gradient descent on the energy landscape has been growing lately, and this paper continues that line of works by thoroughly describing the "next token denoising" task as an energy minimization problem that approaches a Bayes optimal answer.


### Weaknesses
- **Limited scalability and connection to real transformers**. The ideas in the paper are good, but the sandbox in which they test the ideas is small. The experiments are on toy data where the underlying true distribution is known (*how can this work be extended when the true underlying distribution is not known?*), and they only use a single attention head for a single transformer layer. They also do not consider the case where "interesting parameterizations" of $W_{KQ}$ and $W_{PV}$ may be necessary to solve the task at hand (in their experiments, the training tasks cause these parameters to converge almost to identity). Additionally, the update rule they describe discusses only the denoising of the $L+1$ token, when the power of transformers today comes from their parallelizability, allowing them to evolve all tokens in the input simultaneously. Also, as far as I can conclude, all experiments and propositions treat the input tokens as a set of tokens instead of a sequence of tokens with positional information, as is needed for real tasks.

### Summary
I like the paper and I think it is well written. The setting in which they choose to study the model, though quite limited, is appropriate for evaluating the claims in the paper. It would be nice for the authors to include some real-data experiments to make it easier for others to build off the work, but I believe the content in this paper is of sufficient completeness, correctness, novelty, and quality to be accepted to ICML. As always, I am happy to increase my score during the review process if the authors can clarify my questions and concerns

**Questions For Authors:**

Q1: Fig 1 is very helpful, thank you. But it is still unclear what is included in the prompt? Are all blue dots included in the prompt? It seems that the goal of ICL is to project the noisy "test token" ($\tilde{x}$}) back onto the manifold encoded by the context prompt $E$, but I am still not clear if there is any "sequential information" (e.g., token positions $[1,\ldots,N]$) in the prompt?

Q2: Fig3a Case 2 -- why is the bayes optimal predictor have higher loss than the softmax train? Is this "overfitting" to the training data? Also, is there a reason you only test linear attention on the linear manifold case instead of including it in the non-linear and GMM results?

Q3: I find it very interesting that the optimal weights for a single transformer layer/head can be expressed as scaled identity matrices. Could one interpretation of this idea be that the optimal predictor for this task is trying to "attract" the noisy token to the prompt tokens themselves (and not some transformed version of the prompt tokens)? In which case, it is of no surprise that the optimal parameterization leads to identity matrices.

**Relation To Broader Scientific Literature:**

This paper contributes to the growing body of evidence that treats Transformers (specifically, the attention operation of Transformers) as a form of Associative Memory. It also builds on a growing interest in formalizing ICL as a form of memory lookup given the previous tokens in the sequence.

**Theoretical Claims:**

I did not review the proofs in the appendix for correctness, but I saw no glaring issues in the propositions of the the main paper.

---

> ### Author Rebuttal · Authors · 2025-03-31
>
> We sincerely appreciate the reviewer's thoughtful review describing our paper as 'a pleasure to read' with 'clear exposition and motivations.' We're encouraged by the recognition of our 'strongly defended connection of ICL to Dense AM' and we are particularly thankful that the reviewer found the paper to be of 'sufficient completeness, correctness, novelty, and quality to be accepted to ICML.'
>
> **Re: Strengths and Weaknesses**
> We thank the reviewer for their thoughtful feedback on the paper's limitations. While we acknowledge the controlled nature of our experimental setting, this design was deliberate to establish clear theoretical connections and optimal bounds.
>
> - **Synthetic data:** This choice was necessary to derive Bayes-optimal predictors as theoretical baselines. Our findings suggest that with sufficient context, transformers can learn to denoise even when the underlying distribution is not fully known a priori (Fig. 4).
>
> - **Single-layer architecture:** We focused on minimal architectures to isolate fundamental connections between attention and DAM in a novel ICL setting. Our results suggest deeper models might implement multiple gradient steps on a complex energy landscape, and it will be very interesting to explore how this interacts with related but fundamentally distinct efforts including the Energy Transformer (mentioned below by the reviewer.)
>
> - **Identity weights:** While our tasks led to scaled identity weights, we've begun exploring settings with non-isotropic covariance structures that require more complex parameterizations for $W_{KQ}$.
>
> - **Single token denoising:** We deliberately focused on denoising a single token to establish the clearest connection to DAM. Extending to multiple tokens and incorporating sequential information represents important future work.
>
> We view these limitations not as weaknesses of our approach but as exciting avenues for future research that can build upon the theoretical framework we've established.
>
> **Re: References**
> We regret missing the Energy Transformer (ET) citation and are grateful to the reviewer for highlighting it. We will certainly discuss it.
>
> Briefly, while both works explore connections between transformers and DAM networks, their approaches differ fundamentally. Our work examines in-context denoising and demonstrates why a single-step update can be optimal from a Bayesian perspective. In contrast, the ET tackles dataset-specific tasks via multiple iterations using a specialized but elegant design based on Hopfield models. The approaches are complementary but reversed: ET begins with an energy function to construct its architecture, whereas our work shows that standard attention mechanisms naturally learn to perform a gradient step on a context-aware DAM energy landscape. By starting from a vanilla attention layer in a minimal setting -- where true distributions are known and theoretical insights can be gleaned -- our contribution concretely explains why the Ramsauer correspondence involves only one iteration of energy minimization rather than many.
>
> We likewise will cite Ambrogioni alongside Hoover et al. in the revision (thank you!).
>
> **Re: Typo in Appendix E**
> We reviewed and did not find a typo there, but we will change $\mathbb{1}$ to $\mathbb{1}_L$ for notational consistency.
>
> **Re: Questions for Authors**
> - **(Q1)** The blue tokens in Fig. 1 represent the $L$ "pure" tokens sampled from $p_{X}(x)$. These form the context portion of the prompt, with the query being the corrupted $(L+1)^\textrm{th}$ token. Our formulation deliberately focuses on tasks where token order isn't relevant to maximize clarity in the transformer-DAM connection without positional embedding complexities. Extending this to sequence-dependent tasks (e.g. trajectory inference in dynamical systems) represents an exciting direction for future work.
> - **(Q2a)** The apparent outperformance in Fig. 3a (Case 2) represents mild training set overfitting. Test set performance remains bounded by the theoretical optimum.
> - **(Q2b)** While theory indicated softmax attention was appropriate for nonlinear manifold and mixture cases, linear attention also performs well. We can add a supplementary figure showing these additional results.
> - **(Q3)** This interpretation is insightful. The scaled identity weights suggest optimal operation attracts the noisy token toward context tokens themselves rather than transformed versions. This aligns with our finding that the network performs gradient descent on a DAM energy landscape where context tokens serve as stored patterns. We've also explored minimal generalizations where $W_{KQ}$ that are not scaled identity arise from more elaborate covariance structures.
>
> --
>
> We thank the reviewer again for their valuable insights and constructive feedback towards improving the work. We hope our responses have addressed their questions and concerns while further clarifying the contributions of our work.

---

> > ### Comment · Reviewer_PjG9 · 2025-04-03
> >
> > I thank the authors for their response. I believe this is a novel work of high quality that is worthy of acceptance to ICML. I am increasing my score accordingly. Best of luck

---

> > > ### Author Response · Authors · 2025-04-08
> > >
> > > We thank the reviewer again for their positive feedback. We will try our best to incorporate their suggestions in the camera-ready version if accepted.

---

### Decision · Program_Chairs · 2025-05-01

**Decision:**

Accept (oral)

**Comment:**

The paper received strong reviews. In-context denoising task elegantly highlights the connection between Dense Associative Memories (DAM) and attention for tasks extending beyond precise recall of a memorized pattern. This work significantly adds to the growing literature connecting attention operation and DAM networks. Reviewers emphasize correctness, novelty, and high quality of the results. Several reviewers suggested incorporating additional references, which the authors will include in the final camera-ready version. The paper also offers novel perspectives on the one step energy-decent dynamics in DAMs. I share reviewers’ enthusiasm about this work and recommend acceptance.